# Risk-Aware Reinforcement Learning with Coherent Risk Measures and Non-linear Function Approximation

**Thanh Lam**[†]     **Arun Verma**[†]     **Bryan Kian Hsiang Low**[†]     **Patrick Jaillet**[‡]

[†]Department of Computer Science, National University of Singapore, Republic of Singapore
[‡]Department of Electrical Engineering and Computer Science, MIT, USA
{chithanh, arun, lowkh}@comp.nus.edu.sg[†]     jaillet@mit.edu[‡]

## Abstract

We study the risk-aware reinforcement learning (RL) problem in the episodic finite-horizon Markov decision process with unknown transition and reward functions. In contrast to the risk-neutral RL problem, we consider minimizing the risk of having low rewards, which arise due to the intrinsic randomness of the MDPs and imperfect knowledge of the model. Our work provides a unified framework to analyze the regret of risk-aware RL policy with *coherent risk measures* in conjunction with non-linear function approximation, which gives the first sub-linear regret bounds in the setting. Finally, we validate our theoretical results via empirical experiments on synthetic and real-world data.

## 1 Introduction

Reinforcement learning (RL) (Sutton & Barto, 2018) is a control-theoretic problem in which an agent interacts with an unknown environment and aims to maximize its expected total reward. Due to the intrinsic randomness of the environment, even a policy with high expected total rewards may occasionally produce very low rewards. This uncertainty is problematic in many real-life applications like competitive games (Mnih et al., 2013) and healthcare (Liu et al., 2020), where the agent (or decision-maker) needs to be risk-averse. For example, the drug responses to patients are stochastic due to the patients' varying physiology or genetic profiles (McMahon & Insel, 2012); therefore, it is desirable to select a set of treatments that yield high effectiveness and minimize the possibility of adverse effects (Beutler et al., 2016; Fatemi et al., 2021). The existing RL policies that maximize the risk-neutral total reward can not lead to an optimal risk-aware RL policy for problems where the total reward has uncertainty (Yu et al., 2018). Therefore, our goal is to design an RL algorithm that learns a risk-aware RL policy to minimize the risk of having a small expected total reward.

*Then, how should we learn a risk-aware RL policy?* A natural approach is to directly learn a risk-aware RL policy that minimizes the risk of having a small expected total reward (Howard & Matheson, 1972). For quantifying such a risk, one can use risk measures like entropic risk (Föllmer & Knispel, 2011), value-at-risk (VaR) (Dempster, 2002), conditional value-at-risk (CVaR) (Rockafellar et al., 2000), or entropic value-at-risk (EVaR) (Ahmadi-Javid, 2012). These risk measures capture the total reward volatility and quantify the possibility of rare but catastrophic events. The entropic risk measure can be viewed as a mean-variance criterion, where the risk is expressed as the variance of total reward (Fei et al., 2021). Alternatively, VaR, CVaR, and EVaR use quantile criteria, which are often preferable for better risk management over the mean-variance criterion (Chapter 3 of Kisiala (2015)). Among these risk measures, coherent risk measures[1] such as CVaR and EVaR are preferred as they enjoy compelling theoretical properties such as coherence (Rockafellar et al., 2000).

The risk-aware RL algorithms with CVaR as a risk measure (Bäuerle & Ott, 2011; Yu et al., 2018; Rigter et al., 2021) exist in the literature. However, apart from being customized only for CVaR, these algorithms suffer *two significant shortcomings*. First, most of them focus on the *tabular MDP* setting and need multiple complete traversals of the state space (Bäuerle & Ott, 2011; Rigter et al.,

---

[1]Apart from CVaR and EVaR, the risk measures like g-entropic risk measures, Tail value-at-risk, Proportional Hazard (PH) risk measure, Wang risk measure, and Superhedging price also belong to the coherent risk family. More details about various coherent risk measures are given in Appendix C.

2021). These traversals are prohibitively expensive for problems with large state space and impossible for problems with continuous state space, thus limiting these algorithms' applicability in practice. Second, the existing algorithms considering continuous or infinite state space assume that *MDP is known*, i.e., the probability transitions and reward of each state are known a priori to the algorithm. In such settings, the agent does not need to explore or generalize to unseen scenarios. Therefore, the problem considered in Yu et al. (2018) is a planning problem rather than a learning problem. This paper alleviates both shortcomings by proposing a new risk-aware RL algorithm where MDPs are unknown and uses non-linear function approximation for addressing continuous state space.

Recent works (Jin et al., 2020; Yang et al., 2020) have proposed RL algorithms with function approximation and finite-sample regret guarantees, but they only focus on the risk-neutral RL setting. Extending their results to a risk-aware RL setting is non-trivial due to *two major challenges*. First, the existing analyses heavily rely on the linearity of the expectation in the risk-neutral Bellman equation. This linearity property does not hold in the risk-aware RL setting when a coherent risk measure replaces the expectation in the Bellman equation. *Then, how can we address this challenge?* We overcome this challenge by the non-trivial application of the super-additivity property[2] of coherent risk measures (see Lemma 3 and its application in Appendix 4).

The risk-neutral RL algorithms only need one sample of the next state to construct an unbiased estimate of the Bellman update (Yang et al., 2020) as one can unbiasedly estimate the expectation in the risk-neutral Bellman equation with a single sample. However, this does not hold in the risk-aware RL setting. Furthermore, whether one can construct an unbiased estimate of an arbitrary risk measure using only one sample is unknown. This problem leads to the second major challenge: *how can we construct an unbiased estimate of the risk-aware Bellman update?* To resolve this challenge, we assume access to a *weak simulator*[3] that can sample different next states given the current state and action and use these samples to construct an unbiased estimator. Such an assumption is mild and holds in many real-world applications, e.g., a player can anticipate the opponent's next moves and hence the possible next states of the game. After resolving both challenges, we propose an algorithm that uses a risk-aware value iteration procedure based on the upper confidence bound (UCB) and has a finite-sample sub-linear regret upper bound. Specifically, our contributions are as follows:

- We first formalize the risk-aware RL setting with coherent risk measures, namely the risk-aware objective function and the risk-aware Bellman equation in Section 3. We then introduce the notion of regret for a risk-aware RL policy.

- We propose a general risk-aware RL algorithm named Risk-Aware Upper Confidence Bound (RA-UCB) for an entire class of *coherent risk measures* in Section 4. RA-UCB uses UCB-based value functions with non-linear function approximation and also enjoys a finite-sample sub-linear regret upper bound guarantee.

- We provide a unified framework to analyze regret for any coherent risk measure in Section 4.1. The novelty in our analysis is in the decomposition of risk-aware RL policy's regret by the super-additivity property of coherent risk measures (shown in the proof of Lemma 4 in Appendix D.2).

- Our empirical experiments on synthetic and real datasets validate the different performance aspects of our proposed algorithm in Section 5.

## 1.1 RELATED WORK

Risk-aware MDPs first introduced in the seminal work of Howard & Matheson (1972) with the use of an exponential utility function known as the entropic risk measure. Since then, the risk-aware MDPs have been studied with different risk criteria: optimizing moments of the total reward (Jaquette, 1973), exponential utility or entropic risk (Borkar, 2001; 2002; Bäuerle & Rieder, 2014; Fei et al., 2020; 2021; Moharrami et al., 2022), mean-variance criterion (Sobel, 1982; Li & Ng, 2000; La & Ghavamzadeh, 2013; Tamar et al., 2016), and conditional value-at-risk (Boda & Filar, 2006; Artzner et al., 2007; Bäuerle & Mundt, 2009; Bäuerle & Ott, 2011; Tamar et al., 2015; Yu et al., 2018; Rigter et al., 2021). Vadori et al. (2020) focuses on the variability or uncertainty of the rewards.

---

[2]Super-additivity in the reward maximization setting becomes sub-additivity in the cost minimization setting.

[3]Note that the weak simulator can only sample possible next states and returns no information regarding the rewards. In this sense, our simulator is weaker than the archetypal simulators often assumed in the RL literature.

Many of these existing works assume the MDPs are known a priori (known reward and transition kernels) (Yu et al., 2018), focus on the optimization problem (Bäuerle & Ott, 2011; Yu et al., 2018) or asymptotic behaviors of algorithms (e.g., does an optimal policy exist, and if so, is it Markovian, etc.) (Bäuerle & Ott, 2011; Bäuerle & Rieder, 2014). The closest works to ours are Fei et al. (2021); Fei & Xu (2022), which consider the risk-aware reinforcement learning in the function approximation and regret minimization setting. However, they use the entropic risk measure. In contrast, our work considers a significantly different family of risk measures, namely the coherent risk measures. They are preferable and widely used for risk management (Kisiala, 2015). The analysis in Fei et al. (2021); Fei & Xu (2022) utilizes a technique called exponentiated Bellman equation, which is uniquely applicable to the entropic risk measure (or more generally the exponential utility family) and cannot be readily extended to coherent risk measures. Therefore, our analysis differs significantly from that in Fei et al. (2021); Fei & Xu (2022). Tamar et al. (2015) proposes an actor-critic algorithm for the entire class of coherent risk measures but does not provide any theoretical analysis of the regret.

Safe RL and constrained MDPs represent a parallel approach to obtaining risk-aware policies in the presence of uncertainty. Unlike risk-aware MDPs, safe RL does not modify the optimality criteria. Instead, the risk-aversion is captured via constraints on the rewards or risks (Chow & Pavone, 2013; Chow et al., 2017), or as chance constraints (Ono et al., 2015; Chow et al., 2017). Compared with risk-aware MDPs, the constrained MDPs approach enjoys less compelling theoretical properties. The existence of a global optimal Markov policy using the constrained MDPs is unknown, and many existing algorithms only return locally optimal Markov policies using gradient-based techniques. It makes these methods extremely susceptible to policy initialization (Chow et al., 2017), and hence the best theoretical result one can get in this setting is convergence to a locally optimal policy (Chow et al., 2017). In contrast, our result in this paper considers the regret (or sub-optimality) with respect to the *global optimal* policy.

Distributional RL (Bellemare et al., 2022) attempts to model the state-value distribution, and any risk measure can be characterized by such distribution. Therefore, distributional RL represents a more ambitious approach in which the agent needs to estimate the entire value distribution. Existing distributional RL algorithms need to make additional distributional assumptions to work with distributional estimates such as quantiles (Dabney et al., 2018) or empirical distributions (Rowland et al., 2018). In contrast, our risk-aware RL framework only considers the risk measures that apply to the random state-value. As a trade-off, the demand for data and computational resources to estimate the value distribution at every state can be prohibitively expensive for even moderate-sized problems. We establish more detailed connections between risk-aware RL and distribution RL in Appendix A.

## 2 COHERENT RISK MEASURES

Let $Z \in L^1(\Omega, \mathcal{F}, \mathbb{P})$[4] be a real-valued random variable with a finite mean and the cumulative distribution function $F_Z(z) = \mathbb{P}(Z \leq z)$. For $Z' \in L^1(\Omega, \mathcal{F}, \mathbb{P})$, a function $\rho : L^1(\Omega, \mathcal{F}, \mathbb{P}) \to \mathbb{R} \cup \{+\infty\}$ is a coherent risk measure if it satisfies the following properties:

1. Normalized: $\rho(0) = 0$.
2. Monotonic: If $\mathbb{P}(Z \leq Z') = 1$, then $\rho(Z) \leq \rho(Z')$.
3. Super-additive: $\rho(Z + Z') \geq \rho(Z) + \rho(Z')$.
4. Positively homogeneous: For $\alpha \geq 0$, we have $\rho(\alpha Z) = \alpha \rho(Z)$.
5. Translation invariant: For a constant variable $A$ with value $a$, we have $\rho(Z + A) = \rho(Z) + a$.

Since our reward maximization setting contrasts with the cost minimization setting often considered in the literature, we aim to maximize the risk applied to the random reward, i.e., maximizing $\rho(Z)$. Consequently, the properties of risk measure are upended compared to those usually presented in cost minimization setting (Föllmer & Schied, 2010). For example, super-additivity in the reward maximization setting becomes sub-additivity in the cost minimization setting.

**Empirical estimation of the risk.** The risk of a random variable $\rho(Z)$ is completely determined by the distribution of $Z$ ($F_Z$). In practice, we do not know the distribution $F_Z$; instead, we can observe

---

[4]In our risk-aware RL setting, the random variable $Z$ represents the random total reward of the agent.

$m$ independent and identically distributed (IID) samples $\{Z_i\}_{i=1}^m$ from the distribution $F_Z$. Then we can use these samples to get an empirical estimator of $\rho(Z)$, which is denoted by $\hat{\rho}(Z_1, \dots, Z_m)$.

## 3    PROBLEM SETTING

We consider an episodic finite-horizon Markov decision process (MDP), denoted by a tuple $\mathcal{M} = (\mathcal{S}, \mathcal{A}, H, \mathbb{P}, r)$, where $\mathcal{S}$ and $\mathcal{A}$ are sets of possible states and actions, respectively, $H \in \mathbb{Z}_+$ is the episode length, $\mathbb{P} = \{\mathbb{P}_h\}_{h \in [H]}$ are the state transition probability measures, and $r = \{r_h : \mathcal{S} \times \mathcal{A} \to [0,1]\}_{h \in [H]}$ : are the deterministic reward functions. We assume $\mathcal{S}$ is a measurable space of possibly infinite cardinality, and $\mathcal{A}$ is a finite set. For each $h \in [H]$, $\mathbb{P}_h(\cdot|x, a)$ denotes the probability transition kernel when the agent takes action $a$ at state $x$ in time step $h$.

An agent interacts with the MDP as follows. There are $T$ episodes. In the $t$-th episode, the agent begins at state $x_1^t$ chosen arbitrarily by the environment. In each step $h \in [H]$, the agent observes a state $x_h^t \in \mathcal{S}$, selects an action $a_h^t \in \mathcal{A}$, and receives a reward $r_h(x_h^t, a_h^t)$. The MDP then transitions to the next state following the probability transition kernel $x_{h+1}^t \sim \mathbb{P}_h(\cdot|x_h^t, a_h^t)$. The episode terminates when the agent reaches state $x_{H+1}$ at time step $H + 1$. In the last time step, the agent takes no action and receives no reward.

A policy $\pi$ of an agent is a sequence of $H$ functions, i.e., $\pi = \{\pi_h\}_{h \in [H]}$, in which each $\pi_h(\cdot|x)$ is a probability distribution over $\mathcal{A}$. Here, $\pi_h(a|x)$ indicates the probability that the agent takes action $a$ at state $x$ in time step $h$. Any policy $\pi$ and an initial state $x_1$ determine a probability measure $P_{x_1}^\pi$ and an associated stochastic process $\{(x_h, a_h), h \in [H]\}$. Let $\mathbb{E}_{x_1}^\pi[\cdot]$ denote the expectation operator with respect to $P_{x_1}^\pi$. The standard risk-neutral MDP objective is

$$\max_\pi \mathbb{E}_{x_1}^\pi \left[ \sum_{h=1}^H r_h(x_h, a_h) \right]. \tag{1}$$

### 3.1    RISK-AWARE EPISODIC MDP

The risk-neutral objective defined in Eq. (1) does not account for the risk incurred due to the stochasticity in the state transitions and the agent's policy. Markov risk measures (Ruszczyński, 2010) are proposed to model and analyze such risks. The risk-aware MDP objective is defined as

$$\max_\pi J^\pi(x_1), \quad \text{where} \quad J^\pi(x_1) := r_1(x_1, a_1) + \rho(r_2(x_2, a_2) + \rho(r_3(x_3, a_3) + \dots)), \tag{2}$$

where $\rho$ is a coherent one-step conditional risk measure (Ruszczyński, 2010, Definition 6), and $\{x_1, a_1, x_2, a_2, \dots\}$ is a trajectory of states and actions from the MDP under policy $\pi$. Here, $J^\pi$ is defined as a nested and multi-stage composition of $\rho$, rather than through a single-stage risk measure on the cumulative reward $\rho\left(\sum_{h=1}^H r_h(x_h, a_h)\right)$.

The choice of the risk-aware objective function in Eq. (2) has two advantages. Firstly, it guarantees the existence of an optimal policy, and furthermore, this optimal policy is Markovian. Please refer to Theorem 4 in Ruszczyński (2010) for a rigorous treatment of the existence of the optimal Markov policy. Secondly, the above risk-aware objective satisfies the *time consistency* property. This property ensures that we do not contradict ourselves in our risk evaluation. The sequence that is better today should continue to be better tomorrow, i.e., our risk preference stays the same over time. Note that in standard RL, where the risk measure is replaced with expectation, this property is trivially satisfied. In contrast, a single-stage risk measure (i.e., static version) applied on the cumulative reward $\rho\left(\sum_{h=1}^H r_h(x_h, a_h)\right)$ does not enjoy this time consistency property (Ruszczyński, 2010). More detailed discussions about this are in Appendix B.

### 3.2    BELLMAN EQUATION AND REGRET

The risk-aware Bellman equation is developed for the risk-aware objective defined in Eq. (2) (Ruszczyński, 2010). More specifically, let us define the risk-aware state- and action-value functions with respect to the Markov risk measure $\rho$ as

$$V_h^\pi(x) = r_h(x, \pi_h(x)) + \rho\Big(r_{h+1}(x_{h+1}, \pi_{h+1}(x_{h+1})) + \rho\big(r_{h+2}(x_{h+2}, \pi_{h+2}(x_{h+2})) + \dots\big)\Big),$$

$$Q_h^\pi(x, a) = r_h(x, a) + \rho\Big(r_{h+1}(x_{h+1}, \pi_{h+1}(x_{h+1})) + \rho\big(r_{h+2}(x_{h+2}, \pi_{h+2}(x_{h+2})) + \dots\big)\Big).$$

We also define the optimal policy $\pi^\star$ to be the policy that yields the optimal value function $V_h^\star(x) = \sup_\pi V_h^\pi(x)$. The advantage of the formulation given in Eq. (2) is that one can show that the optimal policy exists, and it is Markovian (Theorem 4 of Ruszczyński (2010)). For notations convenience, for any measurable function $V : \mathcal{S} \to [0, H]$, we define the operator $D_h^\rho$ as

$$(D_h^\rho V)(x, a) \coloneqq \rho\left(V(x')\right), \tag{3}$$

where the risk measure $\rho$ is taken over the random variable $x' \sim \mathbb{P}_h(\cdot|x, a)$. Then, the risk-aware Bellman equation associated with a policy $\pi$ takes the form

$$Q_h^\pi(x, a) = (r_h + D_h^\rho V_{h+1}^\pi)(x, a), \quad V_h^\pi(x) = \langle Q_h^\pi(x, \cdot), \pi_h(\cdot|x)\rangle_\mathcal{A}, \quad V_{H+1}^\pi(x) = 0,$$

where $\langle \cdot, \cdot \rangle_\mathcal{A}$ denote the inner product[5] over $\mathcal{A}$ and $(f + g)(x) = f(x) + g(x)$ for function $f$ and $g$. Similarly, the Bellman optimality equation is given by

$$Q_h^\star(x, a) = (r_h + D_h^\rho V_{h+1}^\star)(x, a), \quad V_h^\star(x) = \max_{a \in \mathcal{A}} Q_h^\star(x, a), \quad V_{H+1}^\star(x) = 0. \tag{4}$$

The above equation implies that the optimal policy $\pi^\star$ is the greedy policy with respect to the optimal action-value function $\{Q_h^\star\}_{h \in [H]}$.

In the episodic MDP setting, the agent interacts with the environment through $T$ episodes to learn the optimal policy. At the beginning of episode $t$, the agent selects a policy $\pi^t$, and the environment chooses an initial state $x_1^t$. The difference in values between $V_1^{\pi^t}(x_1^t)$ and $V^\star(x_1^t)$ quantifies the sub-optimality of $\pi^t$, which serves as the regret of the agent at episode $t$. The total regret after $T$ episodes is defined as

$$\mathfrak{R}_T(\rho) = \sum_{t=1}^{T} \left[ V_1^\star(x_1^t) - V_1^{\pi_t}(x_1^t) \right]. \tag{5}$$

We use the widely adopted notion of regret in the risk-neutral setting (Jin et al., 2020; Yang et al., 2020) and risk-aware setting (Fei et al., 2020; 2021). Here, the policy's regret depends on the risk measure $\rho$ via the optimal policy $\pi^\star$. A good policy should have sub-linear regret, i.e., $\lim_{T \to \infty} \mathfrak{R}_T/T = 0$, which implies that the policy will eventually learn to select the best risk-averse actions.

**Remark 1.** Given two risk measures $\rho_1$ and $\rho_2$ with $\mathfrak{R}_T(\rho_1) < \mathfrak{R}_T(\rho_2)$, does not imply $\rho_1$ is a better choice of risk measure for the given problem. Because the optimal policies for $\rho_1$ and $\rho_2$ can be different, their regrets are not directly comparable. Therefore, we cannot use regret as a measure to compare or select the risk measure.

### 3.3 WEAK SIMULATOR ASSUMPTION

One key challenge for the risk-aware RL policy is that the empirical estimation of risk is more complex than the estimation of expectation in risk-neutral RL (Yu et al., 2018). In this paper, we assume the existence of a weak simulator that we can use to draw samples from the probability transition kernel $P_h(\cdot|x, a)$ for any $h \in [H], x \in \mathcal{S}, a \in \mathcal{A}$. This assumption is much weaker than the archetypal simulator assumptions often seen in the RL literature, as they also allow to query reward of a given state and action $r_h(x, a)$. To the best of our knowledge, all existing works in risk-aware RL with coherent risk measures require some assumptions on the transition probabilities to facilitate the risk estimation procedure. Among these assumptions, our weak simulator assumption is the weakest.

### 3.4 ESTIMATING NON-LINEAR FUNCTIONS

We use reproducing kernel Hilbert space (RKHS) as the class of non-linear functions to represent the optimal action-value function $Q_h^*$. For notational convenience, let us denote $z = (x, a)$ and $\mathcal{Z} = \mathcal{S} \times \mathcal{A}$. Following the standard setting, we assume that $\mathcal{Z}$ is a compact subset of $\mathbb{R}^d$ for fixed dimension $d$. Let $\mathcal{H}$ denote the RKHS defined on $\mathcal{Z}$ with the kernel function $k : \mathcal{Z} \times \mathcal{Z} \to \mathbb{R}$. Let $\langle \cdot, \cdot \rangle_\mathcal{H}$ and $\|\cdot\|_\mathcal{H}$ be the inner product and the RKHS norm on $\mathcal{H}$, respectively. Since $\mathcal{H}$ is an RKHS, there exists a feature map $\phi : \mathcal{Z} \to \mathcal{H}$ such that $\phi(z) = k(z, \cdot)$ and $f(z) = \langle \phi(z), f\rangle_\mathcal{H}$ for all $f \in \mathcal{H}$ and for all $z \in \mathcal{Z}$, this is known as the reproducing kernel property.

---

[5]Since $\mathcal{A}$ is a finite set, the inner product over $\mathcal{A}$ is the canonical inner product on Euclidean vector space.

## 4 RISK-AWARE RL ALGORITHM WITH COHERENT RISK MEASURES

We now introduce our algorithm named *Risk-Aware Upper Confidence Bound* (RA-UCB), which is built upon the celebrated *Value Iteration Algorithm* (Sutton & Barto, 2018). RA-UCB first estimates the value function using kernel least-square regression. Then, it computes an optimistic bonus that gets added to the estimated value function to encourage exploration. Finally, it executes the greedy policy with respect to the estimated value function in the next episode.

---

**RA-UCB R**isk-**A**ware **U**pper **C**onfidence **B**ound

---

**Input:** Hyperparameters of coherent risk measure $\rho$ (e.g., confidence level $\alpha \in (0, 1)$ for CVaR)
2: **for** episode $t = 1, 2, \ldots, T$ **do**
3:     Receive the initial state $x_1^t$ and initialize $V_{H+1}^t$ as the zero function.
4:     **for** step $h = H, \ldots, 1$ **do**
5:         For $\tau \in [t-1]$, draw $m$ samples from the weak simulator and construct the response vector $y_h^t$ using Eq. (7).
6:         Compute $\mu_h^t$ and $\sigma_h^t$ using Eq. (8).
7:         Compute $Q_h^t$ and $V_h^t$ using Eq. (9).
8:     **end for**
9:     **for** step $h = 1, \ldots, H$ **do**
10:         Take action $a_h^t \leftarrow \underset{a \in \mathcal{A}}{\arg\max}\, Q_h^t(x_h^t, a)$.
11:         Observe reward $r_h(x_h^t, a_h^t)$ and the next state $x_{h+1}^t$.
12:     **end for**
13: **end for**

---

Recall that we defined $z = (x, a)$ and $\mathcal{Z} = \mathcal{S} \times \mathcal{A}$ in Section 3.4. We define the following Gram matrix $K_h^t \in \mathbb{R}^{(t-1) \times (t-1)}$ and a function $k_h^t : \mathcal{Z} \to \mathbb{R}^{t-1}$ associated with the RKHS $\mathcal{H}$ as

$$K_h^t = \left[ k(z_h^\tau, z_h^{\tau'}) \right]_{\tau, \tau' \in [t-1]}, \quad k_h^t(z) = \left[ k(z_h^1, z), \ldots, k(z_h^{t-1}, z) \right]^\top. \tag{6}$$

Given the observed histories and the weak simulator, we define the response vector $y_h^t \in \mathbb{R}^{t-1}$ as

$$[y_h^t] = \left[ r_h(x_h^\tau, a_h^\tau) + \hat{\rho}(\{V_{h+1}^t(x'_{(i)})\}_{i=1}^m) \right]_{\tau \in [t-1]}, \tag{7}$$

where $\{x'_{(i)}\}_{i=1}^m$ are $m$ next states drawn from the weak simulator $P_h(\cdot | x_h^\tau, a_h^\tau)$. This step contains one of the key differences between RA-UCB and its risk-neutral counterpart, with the presence of the empirical risk estimator in the definition of the response vector $y_h^t$. With the newly introduced notations, we define two functions $\mu_t : \mathcal{Z} \to \mathbb{R}$ and $\sigma_t : \mathcal{Z} \to \mathbb{R}$ as

$$\mu_h^t(z) = k_h^t(z)^\top (K_h^t + \lambda \cdot I)^{-1} y_h^t,$$
$$\sigma_h^t(z) = \lambda^{-1/2} \cdot \left[ k(z, z) - k_h^t(z)^\top (K_h^t + \lambda I)^{-1} k_h^t(z) \right]^{1/2}. \tag{8}$$

The terms $\mu_h^t$ and $\sigma_h^t$ have several important connections with other literature. More specifically, it resembles the posterior mean and standard deviation of a Gaussian process regression problem (Rasmussen, 2003), with $y_h^t$ as its target. The second term $\sigma_h^t$ also reduces to the UCB term used in linear bandits when the feature map $\phi$ is finite-dimensional (Lattimore & Szepesvári, 2020). We then define our estimate of the value functions $Q_h^t$ and $V_h^t$ as follows:

$$Q_h^t(x, a) := \min \left\{ \mu_h^t(x, a) + \beta \cdot \sigma_h^t(x, a), H - h + 1 \right\}, \quad V_h^t(x) := \max_{a \in \mathcal{A}} Q_h^t(x, a), \tag{9}$$

where $\beta > 0$ is an exploration versus exploitation trade-off parameter.

To get some insights on the algorithm, notice that Eq. (7) implements the one-step Bellman optimality update in Eq. (4). To see this, let $X' \sim P_h(\cdot | x_h^\tau, a_h^\tau)$ be the random variable representing the next state. Recall that $V_{h+1}^t$ is the estimated value function by our algorithm at episode $t$. Thus, $V_{h+1}^t(X')$ is also a random variable, where the randomness comes from $X'$. Here, we can start looking at $\rho(V_{h+1}^t(X'))$, i.e., the risk measure $\rho$ applied on the random variable $V_{h+1}^t(X')$. Intuitively, this can be interpreted as the *risk-adjusted value of the next state*. The second term in Eq. (7) above, $\widehat{\rho}(\{V_{h+1}^t(x'_{(i)})\}_{i=1}^m)$, is an empirical estimate of $\rho(V_{h+1}^t(X'))$.

The choice of the response vector in Eq. (7) represents the primary novelty in our algorithm design. This choice enables a new regret decomposition and an upper bound using the concentration inequality of the risk estimator. More details are presented in Appendix D.1.

## 4.1 MAIN THEORETICAL RESULTS

This section presents our main theoretical result, i.e., the regret upper bound guarantee of RA-UCB. We first outline the key assumption that enables the efficient approximation of the value function.

**Assumption 1.** Let $R > 0$ be a fixed constant, $\mathcal{H}$ be the RKHS, and $\mathcal{B}(r) = \{f \in \mathcal{H} : \|f\|_{\mathcal{H}} \leq r\}$ to be the RKHS-norm ball with radius $r$. We assume that for any $h \in [H]$ and any $Q : \mathcal{S} \times \mathcal{A} \to [0, H]$, we have $\mathbb{T}_h^* Q \in \mathcal{B}(RH)$, where $\mathbb{T}_h^*$ is the Bellman optimality operator defined in Eq. (4).

This assumption postulates that the risk-aware Bellman optimality operator maps any bounded action-value function to a function in an RKHS $\mathcal{H}$ with a bounded norm. This assumption ensures that for all $h \in [H]$, the optimal action-value function $Q_h^*$ lies inside $\mathcal{B}(RH)$. Consequently, there is no approximation error when using functions from $\mathcal{H}$ to approximate $Q_h^*$. It can be viewed as equivalent to the *realizability* assumption in supervised learning. Similar assumptions are made in Jin et al. (2020); Yang et al. (2020); Zanette et al. (2020). Please refer to Du et al. (2019) for a discussion on the necessity of this assumption.

Given this assumption, it is clear that the complexity of $\mathcal{H}$ plays a central role in the regret bound of RA-UCB. Following the seminal work of Srinivas et al. (2009), we characterize the intrinsic complexity of $\mathcal{H}$ with the notion of maximum information gain defined as

$$\Gamma_k(T, \lambda) = 1/2 \sup_{\mathcal{D} \subseteq \mathcal{Z}, |\mathcal{D}| \leq T} \{\log \det(I + K_{\mathcal{D}}/\lambda)\}, \tag{10}$$

where $k$ is the kernel function, $\lambda > 0$ is a parameter, and $K_{\mathcal{D}}$ is the Gram matrix. The maximum information gain depends on how fast the eigenvalues of $\mathcal{H}$ decay to zero and can be viewed as a proxy for the dimension of $\mathcal{H}$ when $\mathcal{H}$ is infinite-dimensional. Note that $\Gamma_k(T, \lambda)$ is a problem-dependent quantity that depends on the kernel $k$, state space $\mathcal{S}$, and action space $\mathcal{A}$. Furthermore, let us first define the action-value function classes $\mathcal{Q}_{\text{ucb}}(h, R, B)$ as

$$\mathcal{Q}_{\text{ucb}}(h, R, B) = \{Q :$$
$$Q(z) = \min\{f(z) + \beta \cdot \lambda^{-1/2}[k(z, z) - k_{\mathcal{D}}(z)^{\top}(K_{\mathcal{D}} + \lambda I)^{-1}k_{\mathcal{D}}(z)]^{1/2}, H - h + 1\}^+,$$
$$f \in \mathcal{H}, \|f\|_{\mathcal{H}} \leq R, \beta \in [0, B], |\mathcal{D}| \leq T\}. \tag{11}$$

With the appropriate choice of $R$ and $B$, the set $\mathcal{Q}_{\text{ucb}}(h, R, B)$ contains every possible $Q_h^t$ that can be constructed by RA-UCB. Therefore, the function class $\mathcal{Q}_{\text{ucb}}$ resembles the concept of *hypothesis space* in supervised learning. And as we will see, the complexity of $\mathcal{Q}_{\text{ucb}}$, in particular, the covering number of $\mathcal{Q}_{\text{ucb}}$, plays a crucial role in the regret bound of RA-UCB.

**Theorem 1.** *Let $\lambda = 1 + 1/T$, $\beta = B_T$ in RA-UCB, and let $\Gamma_k(T, \lambda)$ be the maximal information gain defined in Eq. (10). Define a constant $B_T > 0$ that satisfies $B_T = \Theta\big(H(\sqrt{\Gamma_k(T, \lambda)} + \max_{h \in H} \sqrt{\log N_{\infty}(\epsilon, h, B_T)})\big)$. Suppose that the empirical risk estimate $\hat{\rho}$ achieves the rate of $\Xi(m, \delta)$, i.e., $\mathbb{P}\big[|\rho(Z) - \hat{\rho}(\{Z_i\}_{i=1}^m)| \leq \Xi(m, \delta)\big] \geq 1 - \delta$. Then, under Assumption 1, with a probability of at least $1 - (T^2 H^2)^{-1}$, the regret of RA-UCB is*

$$\mathfrak{R}_T \leq 5B_T H \sqrt{T \Gamma_k(T, \lambda)} + 2TH \cdot \Xi\big(m, (8T^3 H^3)^{-1}\big).$$

The proof of Theorem 1 is in Appendix D.1. The regret upper bound consists of two terms. The first term resembles risk-neutral regret bound (Yang et al., 2020, Theorem 4.2). Interestingly, our bound distinguishes itself from the risk-neutral setting with the presence of the second term, which quantifies how fast one can estimate the risk from observed samples. It originates from the risk-aware Bellman optimality equation, in which the one-step update requires knowledge of the risk-to-go starting from the next state (see Eq. (4) for more detail). This risk-to-go quantity is approximated by its empirical counterpart, and the discrepancies give rise to the second term in regret.

Due to the weak simulator assumption, we have good control over the second term. In the following result, we derive the number of samples sufficient to achieve the order-optimal regret for the Conditional Value-at-Risk (CVaR), which is one of the most commonly used coherent risk measures. More details on CVaR and its properties are given in Appendix C.1.

**Corollary 1.** *Let $\rho$ be the CVaR measure defined in Eq. (13) and $\hat{\rho}$ be the CVaR estimator defined in Eq. (14). Then, under the same conditions in Theorem 1, the algorithm RA-UCB achieves the regret of $\mathfrak{R}_T = O\big(B_T H \sqrt{T\Gamma_k(T,\lambda)}\big)$ with*

$$O\Big(TH \cdot \log\big(T^5 H^6 / B_T^2 \Gamma_k(T,\lambda)\big)\Big)$$

*total samples (across all $T$ episodes) from the weak simulator.*

The detailed proof of Corollary 1 is in Appendix D.5. As an example, for the commonly used squared exponential (SE) kernel, we get $B_T = O\big(H \cdot \sqrt{\log(TH)} \cdot (\log T)^d\big)$ (Yang et al., 2020, Corollary 4) and $\Gamma_k(T,\lambda) = O\big((\log T)^{d+1}\big)$ (Srinivas et al., 2009), and thus RA-UCB incurs a regret of $\mathfrak{R}_T = \tilde{O}\big(H^2\sqrt{T}(\log T)^{1.5d+1}\big)$. This result leads to the first sub-linear regret upper bound of the risk-aware RL policy with coherent risk measures.

## 5 EXPERIMENTS

In this section, we empirically demonstrate the effectiveness of RA-UCB. We run different experiments on synthetic and real-world data with the CVaR as a risk measure, which is a commonly used coherent risk measure. We analyze the influence of the risk aversion parameter $\alpha$ (or confidence level for CVaR) on the total reward as well as the behavior of the output policies. The code for these experiments is available in the supplementary material.

### 5.1 SYNTHETIC EXPERIMENT: ROBOT NAVIGATION

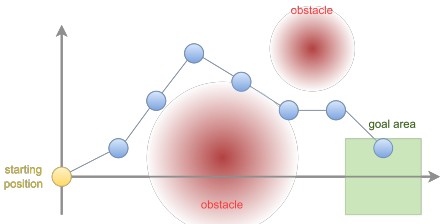

The robot navigation environment is a continuous version of the cliff walking problem considered in example 6.6 of Sutton & Barto (2018), visualized in Fig. 1. In this synthetic experiment, a robot must navigate inside a room full of obstacles to reach its goal destination. The robot navigates by choosing from 4 actions {up, down, left, right}. Since the floor is slippery, the direction of movement is perturbed by $r \cdot \phi$, where $\phi \sim U(-\pi, \pi)$ and $r \in [0,1]$ represent the angle and magnitude of the perturbation. The robot receives a positive reward of 10 for reaching the destination and a negative reward for being close to obstacles. The negative reward increases exponentially as the robot comes close to the obstacle. We set the horizon of each episode to $H = 30$. The robot does not know perturbation parameters ($r = 0.3$) and the obstacles' positions, so it has to learn them online via interacting with the environment. We approximate the state-action value function using the RBF kernel and the `KernelRidge` regressor from Scikit-learn.

Figure 1: Illustration of the continuous version of the cliff walking problem. The robot starts at $(0,0)$ and must navigate to the goal area (in green). The robot gets negative rewards for being close to the obstacles and receives a reward of 10 upon reaching the goal.

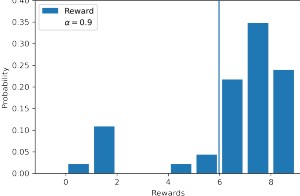
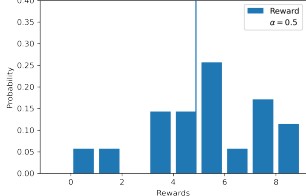
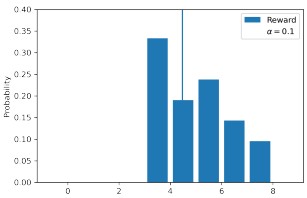

Figure 2: Estimated distribution of the cumulative reward when following the learned policy for different risk parameters. For $\alpha = 0.9$ (leftmost plot), the policy is more risk-tolerant, which causes the average reward to be higher, but occasionally small. As we decrease $\alpha$, the policy becomes more risk-averse, favoring safer paths with smaller average rewards and higher worst-case rewards.

In Fig. 2, we show the histograms of the robot's cumulative rewards that it receives in 50 episodes by following the learned policy with different values of the risk parameter $\alpha \in [0.9, 0.5, 0.1]$. For smaller values of $\alpha$, the learned policy successfully mitigates the tail risk in the distribution, illustrated by the rightmost histogram having the smallest reward of at least 3.0, whereas the reward could go as low as near 0 for the remaining two policies. As we increase $\alpha$, the policy becomes more risk-tolerant, leading to a higher average reward at the expense of occasional bad rewards. In this experiment, we use $m = 100$ samples from the weak simulator to estimate the risk in Eq. (7).

## 5.2 REAL-WORLD EXPERIMENT: TRADING

This trading setup is a generalization of the betting game environment (Bäuerle & Ott, 2011; Rigter et al., 2021). This experiment considers a simplified foreign exchange trading environment based on real historical exchange rates and volumes between EUR and USD in 12 months of 2017. For simplicity, we fixed the trade volume for each hour at 10000. There are two actions in the environment: buy or sell. The state of the environment includes the current position, which is either *long* or *short*, and a vector of *signal features* containing the historical prices and trading volumes over a short period of time. We customize this environment based on the `ForexEnv` in the python package `gym-anytrading`.[6]

In Fig. 3, we show a histogram of the cumulative terminal wealth achieved by the agents in 100 episodes with different risk parameters, plotted in different colors. Similar to the robot experiment, we demonstrate that for a smaller value of $\alpha$, the policy is risk-averse and successfully mitigates the tail of the distribution. This can be seen that the worst-case wealth for $\alpha = 0.1$ (in green) is higher than for $\alpha = 0.5$ (in red) or $\alpha = 0.9$ (in blue). In this experiment, we use $m = 100$ samples from the

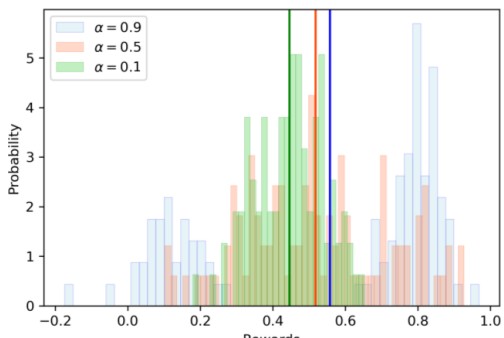

Figure 3: Estimated distribution of the normalized terminal wealth following the learned policy for different risk parameters. The vertical lines represent the average rewards. When $\alpha = 0.9$ (the blue bar), the policy is more risk-tolerant, which causes the average reward to be higher at the expense of occasional low reward. The policy is more risk-averse as we decrease the value of $\alpha$, favoring safe paths with lower average-case rewards and higher worst-case rewards.

weak simulator to estimate the risk in Eq. (7). Additional experiments with other risk measures like VaR and EVaR are given in Appendix E.

**Computational complexity of RA-UCB:** We need to solve $H$ kernel ridge regression problems in each episode. In the $t$-th episode, each regression problem complexity is dominated by two operations: First, the inversion of the Gram matrix $K_h^t$ of size $(t-1) \times (t-1)$ in Eq. (8), which has $O(t^3)$ time complexity and $O(t^2)$ space complexity. Second, the construction of the response vector in Eq. (7) has $O(mt)$ time and space complexity. Therefore, the time and space complexity of the $t$-episode is $O(H(t^3 + mt))$ and $O(H(t^2 + mt))$ respectively.

## 6 CONCLUSION

We proposed a risk-aware RL algorithm named RA-UCB that uses coherent risk measures and non-linear function approximations. We then provided a finite-sample regret upper bound guarantee for RA-UCB and demonstrated its effectiveness in robot navigation and forex trading environments.

The performance of the proposed algorithm depends profoundly on the quality of the empirical risk estimator. This paper assumes access to a weak simulator that can sample the next states, thus effectively alleviating the need to estimate the risk from the observed trajectories. Therefore, a potential future direction is to relax or weaken this assumption, allowing risk-aware RL algorithms to be useful in more practical problems. Another interesting direction is to consider the episodic MDPs, where episodes can have varying lengths horizons or even infinite horizons.

---

[6]The package `gym-anytrading` is available at https://github.com/AminHP/gym-anytrading.

## 7   REPRODUCIBILITY STATEMENT

In this paper, we dedicate a substantial effort to improving the reproducibility and comprehensibility of both our theoretical results and empirical experiments. We formally state and discuss the necessity and implications of our assumptions (please see Section 3.3 and the paragraph below Assumption 1) before presenting our theoretical results. We also provide a 3-step proof sketch of our main theoretical result. For each step, we present the key ideas and high-level directions and refer the reader to more detailed and complete proofs in the Appendices. For the experiments, we provide details of different experimental settings in Section 5, and include our code in the supplementary material.

### ACKNOWLEDGMENTS

This research is part of the programme DesCartes and is supported by the National Research Foundation, Prime Minister's Office, Singapore under its Campus for Research Excellence and Technological Enterprise (CREATE) programme. C. T. Lam is supported by the Singapore-MIT Alliance for Research and Technology (SMART) PhD Fellowship.

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

## A    CONNECTIONS TO DISTRIBUTIONAL RL

We first give a brief survey on the distributional RL literature and discuss its connection to risk-sensitive RL. For a policy $\pi$ in distributional RL, we define the total reward from state $x$ (or state-action pair $(x, a)$) and time step $h$ as the sum of rewards of an agent starting in time $h$, at state $x$ (or state-action pair $(x, a)$), and the following policy $\pi$ as follows

$$G_h^\pi(x) \coloneqq \sum_{h'=h}^{H} r_{h'}(x_{h'}, a_{h'}) | x_h = x, a_{h'} \sim \pi_{h'}(\cdot | x_{h'}), x_{h'+1} \sim \mathbb{P}_{h'}(\cdot | x_{h'}, a_{h'}),$$

$$J_h^\pi(x, a) \coloneqq \sum_{h'=h}^{H} r_{h'}(x_{h'}, a_{h'}) | x_h = x, a_h = a, a_{h'} \sim \pi_{h'}(\cdot | x_{h'}), x_{h'+1} \sim \mathbb{P}_{h'}(x_{h'}, a_{h'}).$$

Both $G_h^\pi(x)$ and $J_h^\pi(x, a)$ are random variables where the randomness comes from the transition probability $\mathbb{P}$ and the (possibly stochastic) policy $\pi$. In standard MDP, the expected value of these random variables, $\mathbb{E}[G^\pi(x)]$ and $\mathbb{E}[J^\pi(x, a)]$, are known as the value function and action-value function, respectively. Distributional RL (Bellemare et al., 2022) is built upon the overarching idea of estimating the value distribution, i.e., the distribution of the random variables $G_h^\pi(x)$ and $J_h^\pi(x, a)$, rather than just the expected values as in standard RL. The most important component of distributional RL is the distributional Bellman equation, which is given as follows:

$$J_h^\pi(x, a) \overset{d}{=} r_h(x, a) + J_h^\pi\big(x', \arg\max_{a' \in \mathcal{A}} \mathbb{E}[J_h^\pi(x', a')]\big),$$

where $\forall h \in [H]$ and $\overset{d}{=}$ denotes equality in distribution, and the next state $x' \sim \mathbb{P}_h(\cdot | x, a)$. Although distributional RL models the entire value distribution, notice that the optimal action is greedy with respect to the expectation of the value of the next state, which resembles that of standard RL.

## B    COHERENT ONE-STEP CONDITIONAL RISK MEASURE

Recall the formulation of the risk-aware MDP objective function given in Eq. (2):

$$\max_{\pi} J^\pi(x_1), \quad \text{where} \quad J^\pi(x_1) \coloneqq r_1(x_1, a_1) + \rho(r_2(x_2, a_2) + \rho(r_3(x_3, a_3) + \dots)),$$

where $\rho$ is a coherent one-step conditional risk measure, and $x_1, a_1, x_2, a_2, \dots$ is a trajectory of states and actions from the MDP under policy $\pi$. Notice that $J^\pi$ is defined as a nested and multi-stage composition of conditional risk measure $\rho$, which is also referred to as the *dynamic risk optimization* problem (Rigter et al., 2021, use CVaR as a risk measure). There are two advantages to this formulation.

Firstly, one can show that the optimal policy exists, and it is Markovian (Theorem 4 in Ruszczyński (2010)). Therefore, we can use the Bellman update to learn the risk-aware RL policy. When the objective function uses a single-stage risk measure on the sum of rewards (i.e., $\rho\left(\sum_{h=1}^{H} r_h(x_h, a_h)\right)$), the problem is referred to as the *static risk optimization*. Under this setting, (Bäuerle & Ott, 2011, also use CVaR as a risk measure) shows that the optimal policy exists but is non-Markovian. Therefore, these history-dependent policies must optimally solve the static risk optimization problem, which is harder than learning a Markovian policy.

Secondly, the above risk-aware objective satisfies the *time consistency* property. Intuitively, the time consistency property indicates that the sequence that is better today should continue to be better tomorrow, i.e., our risk preference stays the same over time. To formally define this concept, let us consider the problem of measuring the risk of sequences.

Let $(\Omega, \mathcal{F}, \mathbb{P})$ denote a probability space with filtration $\mathcal{F}_1 \subset \mathcal{F}_2 \subset \cdots \subset \mathcal{F}_T \subset \mathcal{F}$. Let $Z_1, \dots, Z_T$ denote the adapted sequence of random variables (one may view this as a sequence of random rewards for the purpose of this paper). Finally, we define $\mathcal{Z}_t = \mathcal{L}^p(\Omega, \mathcal{F}_t, \mathbb{P}), p \in [1, \infty)$, and $\mathcal{Z}_{t,T} = \mathcal{Z}_t \times \cdots \times \mathcal{Z}_T$. We define the following notion of *conditional risk measure* and *dynamic risk measure* as follows:

**Definition 1.** A **conditional risk measure** is a mapping $\rho_{t,T} : \mathcal{Z}_{t,T} \to \mathcal{Z}_t$ that satisfies the following monotonicity condition:

$$\rho_{t,T}(Z) \leq \rho_{t,T}(W) \text{ for all } Z, W \in \mathcal{Z}_{t,T} \text{ such that } Z \leq W.$$

A sequence of conditional risk measures: $\{\rho_{t,T}\}_{t=1,\ldots,T}$ is called a **dynamic risk measure**.

One can view the conditional risk measure as a non-linear extension of the conditional expectation. The randomness of the entire sequence $t, \ldots, T$ is reduced to the randomness at time $t$. This is analogous to conditional expectation in which $\mathbb{E}[Z_t + \cdots + Z_T | Z_t]$ is a random variable where the randomness comes from $Z_t$.

Intuitively, $\rho_{t,T}(Z_t, \ldots, Z_T)$ represents the amount of reward a player is willing to take in exchange for a sequence of future random rewards $Z_t + \cdots + Z_T$. For a risk-neutral player, that amount will equal $\mathbb{E}[Z_t + \cdots + Z_T | Z_t]$.

We are now ready to define the notion of time consistency.

**Definition 2.** A dynamic risk measure $\{\rho_{t,T}\}_{t=1,\ldots,T}$ is **time-consistent** if for all $\tau, \theta \in [T]$ and $\tau < \theta$, we have the following: If for all $k = \tau, \ldots, \theta - 1$, $Z_k = W_k$ and $\rho_{\theta,T}(Z_\theta, \ldots, Z_T) \leq \rho_{\theta,T}(W_\theta, \ldots, W_T)$, then we have $\rho_{\tau,T}(Z_\tau, \ldots, Z_T) \leq \rho_{\tau,T}(W_\tau, \ldots, W_T)$.

The time-consistency property implies that a dynamic risk measure ensures that given the same rewards, the sequence that is better today should also be better tomorrow.

Why is time consistency a desirable property? This property ensures that we do not contract ourselves in our risk evaluation. If we observe the same realization, the sequence that is better today should continue to be better tomorrow. Our risk preference stays the same over time. Note that this property is trivially satisfied in standard RL, where the risk measure is replaced with expectation.

In contrast, a single risk measure applied on the cumulative reward $\rho\left(\sum_{h=1}^{H} r_h(x_h, a_h)\right)$ does not enjoy this time consistency property.

## C    COHERENT RISK MEASURES

### C.1    CONDITIONAL VALUE-AT-RISK

Let $Z$ be a finite mean random variable, i.e., $\mathbb{E}[|Z|] < \infty$, with the cumulative distribution function $F_Z(z) = \mathbb{P}(Z \leq z)$ (an example to keep in mind is that $Z$ represents the random total reward of a learning agent). The *value-at-risk* at confidence level $\alpha \in (0, 1)$ is defined as

$$\mathrm{VaR}_\alpha(Z) = \min\{z : F_Z(z) \geq \alpha\}. \tag{12}$$

The minimum is attained because the cumulative distribution function $F_Z$ is a non-decreasing and right-continuous function in $z$. When $F_Z$ is strictly increasing (and thus bijective), $\mathrm{VaR}_\alpha(Z) = F_Z^{-1}(\alpha)$. The *conditional value-at-risk* (also known as the *average value-at-risk*) at confidence level $\alpha \in (0, 1)$ is defined as[7]

$$\mathrm{CVaR}_\alpha(Z) := \frac{1}{\alpha} \int_0^\alpha \mathrm{VaR}_t(Z)dt. \tag{13}$$

If $Z$ is a continuous random variable, then $\mathrm{CVaR}_\alpha(Z) = \mathbb{E}[Z | Z \leq \mathrm{VaR}_\alpha(Z)]$ (Acerbi & Tasche, 2002). From this expression, $\mathrm{CVaR}_\alpha(Z)$ can be viewed as the average of the worst-case $\alpha$-fraction of $Z$. It is easy to see that $\mathrm{CVaR}_1(Z) = \mathbb{E}[Z]$, and as $\alpha \to 0$, $\mathrm{CVaR}_\alpha$ approaches the worst-case (or robust) realization. An important result of CVaR (Rockafellar & Uryasev, 2002, Theorem 10) (also known as the fundamental minimization theorem) is that it can be represented as the solution of a convex optimization problem.

**Lemma 1.** (Keramati et al., 2020; Baudry et al., 2021) *Let $Z$ be a finite mean random variable, and let $\alpha \in (0, 1)$. Then, it holds that*

$$\mathrm{CVaR}_\alpha(Z) = \max_{s \in \mathbb{R}} \left\{ s - \frac{1}{\alpha} \mathbb{E}\left[(s - Z)^+\right] \right\} = \max_{s \in \mathbb{R}} \left\{ s + \frac{1}{\alpha} \mathbb{E}\left[(Z - s)^-\right] \right\},$$

---

[7]Note that the definition of CVaR presented above is different from that in the literature, e.g., in (Rockafellar et al., 2000). This is because we are treating $Z$ as rewards and thus maximizing $Z$, whereas existing works consider $Z$ as costs and consequently minimizing $Z$.

where $(x)^+ = \max\{x, 0\}$ *represents the positive part of x, similarly* $(x)^- = \min\{0, x\}$ *represents the negative part of x, and the maximum point is given by* $s^* = \text{VaR}_\alpha(Z)$.

Conditional Value-at-risk is a prominent risk measure with extensive applications in stochastic optimization (see Rockafellar et al. (2000) for example). By carefully choosing $\alpha$, CVaR can be tuned to be sensitive to rare events with exceptionally low rewards, making it attractive as a risk measure. The CVaR is also known for having favorable mathematical properties such as coherence.

**Empirical estimation of the risk.** Let $Z_1, \dots, Z_m$ be $m$ i.i.d. samples drawn from the distribution $F_Z$, then the empirical estimation of $\text{CVaR}_\alpha(Z)$ is given by

$$\widehat{\text{CVaR}}_\alpha(\{Z_i\}_{i=1}^m) = \max_{s \in \mathbb{R}} \left\{ s + \frac{1}{\alpha m} \sum_{i=1}^m (Z_i - s)^- \right\}. \tag{14}$$

**Lemma 2.** (Lemma 3 in Yu et al. (2018)) *Let* $Z_1, \dots, Z_m \sim F_Z$ *be* $m$ *i.i.d. bounded random variables, i.e.,* $\mathbb{P}[0 \le Z_i \le B] = 1, \forall i$, *then we have*

$$\mathbb{P}\Big[\big|\text{CVaR}_\alpha(Z) - \widehat{\text{CVaR}}_\alpha(\{Z_i\}_{i=1}^m)\big| \ge \varepsilon\Big] \le 2\left(1 + \frac{4}{\varepsilon(1-\alpha)}\right) \exp\left[\frac{-m\varepsilon^2(1-\alpha)^2}{2(2-\alpha)^2 B^2}\right].$$

## C.2 COMPARISONS BETWEEN ENTROPIC RISK MEASURE AND CVAR

The closest work to ours is Fei et al. (2021), which considers the risk-aware RL problem in the function approximation setting. However, they use the entropic risk measure. This section discusses some key differences between the entropic risk measure and CVaR.

**Entropic risk measure.** For a finite-mean random variable $Z$ and a parameter $\beta \ne 0$, the entropic risk measure of $Z$ is defined as

$$\text{ER}_\beta(Z) := \frac{1}{\beta} \log \mathbb{E}[e^{\beta Z}].$$

The entropic risk measure $\text{ER}_\beta$ is the normalized cumulant generating function of $Z$ and is concave and additive for independent random variables (Föllmer & Knispel, 2011). Using Taylor expansion, the entropic risk can be expressed as follows:

$$\text{ER}_\beta(Z) = \mathbb{E}[Z] + \frac{\beta}{2}\text{Var}[Z] + O(\beta^2).$$

From the above expression, we observe that $\beta > 0$ induces a risk-tolerant objective and $\beta < 0$ induces a risk-averse one. As $\beta \to 0$, $\text{ER}_\beta(Z)$ tends to the risk-neutral expectation $\mathbb{E}[Z]$.

However, the additivity property of the entropic risk measure may not be desirable in many practical scenarios. For example, if $\text{ER}_\beta(X_1 + X_2 + \cdots + X_n)$ is the total reward of $n$ i.i.d. random variables, then the reward per random variable is $\text{ER}_\beta(X_1)$, no matter how large $n$ is. Thus, the aggregation of independent risks does not affect 'diversification reduces risks.' In contrast, coherent risk measures like CVaR are super-additive and thus enjoy this property.

## C.3 ENTROPIC VALUE-AT-RISK

Let $(\Omega, \mathcal{F}, P)$ be a probability space. Let $Z$ be a finite mean random variable, i.e., $\mathbb{E}[|Z|] < \infty$, whose moment-generating function $M_Z(z)$ exists for all $z \ge 0$. The *Entropic Value-at-Risk* (EVaR) (Ahmadi-Javid, 2011; 2012) at confidence level $1 - \alpha$ is defined as

$$\text{EVaR}_{1-\alpha}(Z) := \inf_{z > 0} \left\{ z^{-1} \ln\left(\frac{M_Z(z)}{\alpha}\right) \right\}.$$

The EVaR admits a *dual representation* as follows:

$$\text{EVaR}_{1-\alpha}(Z) = \sup_{Q \in \mathcal{Q}} \big[\mathbb{E}_Q(Z)\big],$$

where $\mathcal{Q} = \{Q \ll P : d_{\text{KL}}(Q||P) \le -\ln\alpha\}$ where $d_{\text{KL}}$ denotes the Kullback-Leibler (KL) divergence of $Q$ with respect to $P$. This dual representation of the EVaR also reveals the reason behind its name.

### C.4 G-ENTROPIC RISK MEASURE

Inspired by the dual representation of EVaR, Ahmadi-Javid (2012) proposes a large class of information-theoretic coherent risk measures called *g-entropic* risk measures. This new class contains both the CVaR and EVaR.

Let $g$ be a convex proper function with $g(1) = 0$ and $\beta \geq 0$. The generalized relative entropy of $Q$ with respect to $P$, denoted by $H_g(P, Q)$, is an information-type pseudo-distance (also called divergence measure) from $Q$ to $P$:

$$H_g(P, Q) := \int g\Big(\frac{dQ}{dP}\Big) dP.$$

This quantity is an important divergence measure, initially mentioned in Ali & Silvey (1966); Csiszár (1967), and discussed in more detail in Liese & Vajda (2006); Ullah (1996). For $g(z) = z \ln z$, we obtain the Kullback-Leibler divergence from $Q$ to $P$ (Kullback & Leibler, 1951).

Let $Z$ be a finite-mean random variable. Then, the g-entropic risk measure with divergence level $\beta$ is defined as

$$\text{ER}_{g,\beta} := \sup_{Q \in \mathcal{Q}} \mathbb{E}_Q(Z),$$

where $\mathcal{Q} = \{Q \ll P : H_g(P, Q) \leq \beta\}$. We can show that the CVaR and EVaR are special cases of the g-entropic risk measure, with proper choices of $g$ and $\beta$. For a more comprehensive discussion of the properties of the g-entropic risk measure, please refer to Section 5 of Ahmadi-Javid (2012).

### C.5 OTHER COHERENT RISK MEASURES

Risk measures like Tail value-at-risk, Proportional Hazard (PH) risk measure, Wang risk measure, and Superhedging price also belong to the family of coherent risk measures. These risk measures have many important applications. For example, Proportional Hazard (PH) risk measure is widely used in healthcare domains such as clinical trials (Rulli et al., 2018) or epidemiology (Moolgavkar et al., 2018). Wang risk measure and Superhedging price are commonly used in financial applications such as asset pricing (Wang, 2000) or portfolio optimization (Löhne & Rudloff, 2014).

In the RL context, CVaR is the most well-known and commonly used risk measure among all coherent risk measures and is relatively well-studied in the literature (Bäuerle & Ott, 2011; Yu et al., 2018). Some very recent works (Ni & Lai, 2022a;b) have started investigating the use of EVaR in RL. Unlike our work, the techniques used by Ni & Lai (2022a;b) exploit properties that are unique to EVaR and thus cannot be generalized to others in the family of coherent risk measures.

## D   PROOFS

Before presenting the proofs of the supporting lemmas, we review the properties of coherent risk measures and derive the necessary results needed in our proofs. A coherent risk measure (Föllmer & Schied, 2010) is defined as follows

**Definition 3.** Let $X, Y$ be two random variables. A mapping $\rho$ is called a coherent risk measure if $\rho$ satisfies the following conditions for all $X, Y$:

- Monotonicity: If $X \leq Y$ a.s., then $\rho(X) \leq \rho(Y)$.

- Translation invariance: If $m \in \mathbb{R}$, then $\rho(X + m) = \rho(X) + m$.

- Positive homogeneity: If $\alpha > 0$, then $\rho(\alpha Z) = \alpha \rho(Z)$.

- Super-additivity: $\rho(X + Y) \geq \rho(X) + \rho(Y)$.

We want to highlight in our paper that we consider *maximizing* risks of the *rewards*. It is in direct contrast to other papers that consider *minimizing* risks of the *costs*. Therefore, our properties are upended compared to the properties presented in Föllmer & Schied (2010). The following result presents a simple inequality we will use throughout this section.

**Lemma 3.** *For any two state-action value functions $f_1, f_2 : \mathcal{S} \times \mathcal{A} \to \mathbb{R}$, we have*

$$(D_h^\rho f_1)(x,a) - (D_h^\rho f_2)(x,a) \le -\Big(D_h^\rho\big(-(f_1 - f_2)\big)\Big)(x,a).$$

*Proof.* The super-additivity property of $\rho$ implies that

$$\rho(X) + \rho(Y - X) \le \rho(Y), \quad \text{equivalently,} \quad \rho(X) - \rho(Y) \le -\rho(Y - X). \tag{15}$$

In the inequality given in Eq. (15), we want to highlight the super-additivity properties of $\rho$. We emphasize that the statement $\rho(X) - \rho(Y) \le \rho(X - Y)$ is incorrect since $\rho$ is only *positively* homogeneous. This argument concludes our proof of inequality.

Let $x' \sim \mathbb{P}_h(\cdot|x,a)$ be the random variable represents the next state by following the transition kernel $\mathbb{P}_h$, by definition of $D_h^\rho$ in Eq. (3), we have

$$
\begin{aligned}
(D_h^\rho f_1)(x,a) - (D_h^\rho f_2)(x,a) &= \rho\big(f_1(x')\big) - \rho\big(f_1(x')\big) \\
&\le -\rho\big(-(f_1(x') - f_2(x'))\big) \\
&= -\Big(D_h^\rho\big(-(f_1 - f_2)\big)\Big)(x,a),
\end{aligned}
$$

where the first and last equality follows from the definition of $D_h^\rho$ from Eq. (3), and the inequality is due to Eq. (15). $\qquad\square$

### D.1  PROOF OF THEOREM 1

We first define a few notations to simplify the presentation of the proof. First, we define the temporal-difference (TD) error as

$$\delta_h^t(x,a) = (r_h + D_h^\rho V_{h+1}^t)(x,a) - Q_h^t(x,a), \quad \forall (x,a) \in \mathcal{S} \times \mathcal{A}. \tag{16}$$

For a trajectory $\{(x_h^t, a_h^t)\}_{h\in[H]}$, we further define the two following quantities

$$
\begin{aligned}
\zeta_{t,h}^1 &= \big[V_h^t(x_h^t) - V_h^{\pi_t}(x_h^t)\big] - \big[Q_h^t(x_h^t, a_h^t) - Q_h^{\pi_t}(x_h^t, a_h^t)\big], \\
\zeta_{t,h}^2 &= \big[(D_h^\rho V_{h+1}^t)(x_h^t, a_h^t) - (D_h^\rho V_{h+1}^{\pi_t})(x_h^t, a_h^t)\big] - \big[V_{h+1}^t(x_{h+1}^t) - V_{h+1}^{\pi_t}(x_{h+1}^t)\big]. \tag{17}
\end{aligned}
$$

The random variables $\zeta_{t,h}^1$ and $\zeta_{t,h}^2$ capture the deviations of the value function due to two sources of randomness in the MDP – the randomness of choosing the action $a_h^t \sim \pi_h^t(\cdot|x_h^t)$ and drawing next state $x_{h+1}^t \sim \mathbb{P}_h(\cdot|x_h^t, a_h^t)$. We establish the upper bound in the following steps.

**Step 1: Decomposition of the regret.**

**Lemma 4.** *We can upper bound the regret as*

$$\mathfrak{R}(T) \le \underbrace{-\sum_{t=1}^T \sum_{h=1}^H \Big(\prod_{i=1}^{h-1} \mathbb{J}_{\pi_i^\star} D_i^\rho\Big)\mathbb{J}_{\pi_h^\star}(-\delta_h^t)(x_1^t) - \sum_{t=1}^T \sum_{h=1}^H \delta_h^t(x_h^t, a_h^t)}_{\text{Term I}} + \underbrace{\sum_{t=1}^T \sum_{h=1}^H (\xi_{t,h}^1 + \xi_{t,h}^2)}_{\text{Term II}},$$

*where $\delta_h^t$, $\zeta_{t,h}^1$, and $\zeta_{t,h}^2$ are defined above.*

**Proof sketch.** We decompose the instantaneous regret at the $t$-th episode into

$$V_1^\star(x_1^t) - V_1^{\pi^t}(x_1^t) = \big[V_1^\star(x_1^t) - V_1^t(x_1^t))\big] + \big[V_1^t(x_1^t) - V_1^{\pi^t}(x_1^t)\big].$$

To upper bound the first term, we establish an inequality of the form $V_h^\star - V_h^t \le f(V_{h+1}^\star - V_{h+1}^t)$ for some function $f$ and apply it recursively. This inequality is established using the Bellman equation and the super-additivity property of CVaR. Similar techniques can be applied to the upper bound of the second term. The detailed proof is given in Appendix D.2

**Step 2. Upper bounding Term I.**

**Lemma 5.** *Let $\lambda = 1 + 1/T$ and $\beta = B_T$ in Algorithm RA-UCB. Then under Assumption 1, with probability at least $1 - (2T^2H^2)^{-1}$, we have that for all $t \in [T], h \in [H], x \in \mathcal{S}$, and $a \in \mathcal{A}$:*

$$-2\beta b_h^t(x,a) \le \delta_h^t(x,a) \le 0.$$

The proof of Lemma 5 is in Appendix D.3. By Lemma Lemma 5, $\delta_h^t$ is a negative function, and thus we could upper bound the first term in (I) by 0. We obtain that, with a probability of at least $1 - (2T^2H^2)^{-1}$,

$$\text{Term I} \leq -\sum_{t=1}^{T}\sum_{h=1}^{H} \delta_h^t(x_h^t, a_h^t)$$

$$\leq 2\beta \sum_{t=1}^{T}\sum_{h=1}^{H} b_h^t(x_h^t, a_h^t),$$

which is an upper bound of the sum of the bonus terms. Recall that we can rewrite the bonus term as

$$b_h^t(x_h^t, a_h^t) = \left[\phi(x_h^t, a_h^t)^\top (\Lambda_h^t)^{-1}\phi(x_h^t, a_h^t)\right]^{1/2},$$

where $\Lambda_h^t = \sum_{\tau=1}^{t-1}\phi(x_h^\tau, a_h^\tau)\phi(x_h^\tau, a_h^\tau)^\top + \lambda \cdot I_{\mathcal{H}}$ and $I_{\mathcal{H}}$ is the identity operator on $\mathcal{H}$. Then,

$$\text{Term I} \leq 2\beta \cdot \sqrt{T}\sum_{h=1}^{H}\left[\sum_{t=1}^{T}\phi(x_h^t, a_h^t)^\top(\Lambda_h^t)^{-1}\phi(x_h^t, a_h^t)\right]^{1/2}$$

$$\leq 2\beta \cdot \sqrt{T}\sum_{h=1}^{H}[2\log\det(I + K_h^T/\lambda)]^{1/2}$$

$$= 4\beta H \cdot \sqrt{T \cdot \Gamma_k(T, \lambda)},$$

where $\Gamma_k(T, \lambda)$ is the maximal information gain defined in Eq. (10).

**Step 3. Upper bounding Term II.**

**Lemma 6.** *For $\zeta_{t,h}^1$ and $\zeta_{t,h}^2$ defined in Eq. (17). We have that, with probability at least $1 - \delta$,*

$$\sum_{t=1}^{T}\sum_{h=1}^{H}(\zeta_{t,h}^1 + \zeta_{t,h}^2) \leq \sqrt{16TH^3\log(2/\delta)} + 2TH \cdot \Xi(m, \delta/(4TH)).$$

**Proof sketch.** We show that $\{\zeta_{t,h}^1\}_{(t,h)\in[T]\times[H]}$ is a bounded martingale difference sequence and apply Azuma-Hoeffding concentration inequality. For $\{\zeta_{t,h}^2\}_{(t,h)\in[T]\times[H]}$, we use concentration inequality of the risk estimator. The complete proof is in Appendix D.4

Setting $\delta = (2T^2H^2)^{-1}$ gives us

$$\text{Term II} \leq \sqrt{16TH^3\log(4T^2H^2)} + 2TH \cdot \Xi(m, (8T^3H^3)^{-1})$$

Therefore, combining these above results, with probability at least $1 - (T^2H^2)^{-1}$, the regret is bounded by

$$\mathfrak{R}(T) \leq 4\beta H\sqrt{T\Gamma_k(T, \lambda)} + \sqrt{16TH^3\log(4T^2H^2)} + 2TH \cdot \Xi(m, (8T^3H^3)^{-1})$$

$$\leq 5\beta H\sqrt{T\Gamma_k(T, \lambda)} + 2TH \cdot \Xi(m, (8T^3H^3)^{-1}).$$

Substituting $\beta = B_T$ completes the proof of Theorem 1.

### D.2   PROOF OF LEMMA 4

We decompose the instantaneous regret at the $t$-th episode into

$$V_1^\star(x_1^t) - V_1^{\pi^t}(x_1^t) = \underbrace{V_1^\star(x_1^t) - V_1^t(x_1^t)}_{(A)} + \underbrace{V_1^t(x_1^t) - V_1^{\pi^t}(x_1^t)}_{(B)}.$$

We proceed to upper bound the two terms separately. For ease of presentation, we first define the operator $\mathbb{J}_\pi$ acting on functions $f : \mathcal{S} \times \mathcal{A} \to \mathbb{R}$ that map a state-action value function to the state value function by following policy $\pi$ as follows:

$$(\mathbb{J}_\pi f)(x) = \langle f(x, \cdot), \pi(\cdot|x)\rangle_{\mathcal{A}}.$$

Since the domain of $f(x, \cdot)$ and $\pi(\cdot|x)$ is a finite set $\mathcal{A}$, the inner product above can be interpreted as an inner product between two Euclidean vectors.

**Term (A).** By the definitions of the optimal value function, we have $V_h^\star(x) = \langle Q_h^\star(x, \cdot), \pi_h^\star(\cdot|x)\rangle_\mathcal{A}$ for all $x \in \mathcal{S}$. Similarly, by the definition of $V_h^t$, we get $V_h^t(x) = \langle Q_h^t(x, \cdot), \pi_h^t(\cdot|x)\rangle_\mathcal{A}$ for all $x \in \mathcal{S}$. Thus, for any $t \in [T]$, $h \in [H]$, $x \in \mathcal{S}$, we have

$$
\begin{aligned}
V_h^\star(x) - V_h^t(x) &= \langle Q_h^\star(x, \cdot), \pi_h^\star(\cdot|x)\rangle_\mathcal{A} - \langle Q_h^t(x, \cdot), \pi_h^t(\cdot|x)\rangle_\mathcal{A} \\
&= \langle Q_h^\star(x, \cdot), \pi_h^\star(\cdot|x)\rangle_\mathcal{A} - \langle Q_h^t(x, \cdot), \pi_h^\star(\cdot|x)\rangle_\mathcal{A} + \\
&\quad \langle Q_h^t(x, \cdot), \pi_h^\star(\cdot|x)\rangle_\mathcal{A} - \langle Q_h^t(x, \cdot), \pi_h^t(\cdot|x)\rangle_\mathcal{A} \\
&= \langle Q_h^\star(x, \cdot) - Q_h^t(x, \cdot), \pi_h^\star(\cdot|x)\rangle_\mathcal{A} + \langle Q_h^t(x, \cdot), \pi_h^\star(\cdot|x) - \pi_h^t(\cdot|x)\rangle_\mathcal{A}.
\end{aligned}
$$

Since $\pi_h^t$ is the greedy policy with respect to $Q_h^t$, it gives

$$
\langle Q_h^t(x_h, \cdot), \pi_h^\star(\cdot|x_h) - \pi_h^t(\cdot|x_h)\rangle_\mathcal{A} = \langle Q_h^t(x_h, \cdot), \pi_h^\star(\cdot|x_h)\rangle_\mathcal{A} - \max_{a \in \mathcal{A}} Q_h^t(x_h, a) \leq 0,
$$

for all $x_h \in \mathcal{S}$. As a result, we can upper bound the second term by 0 and have

$$
\begin{aligned}
V_h^\star(x) - V_h^t(x) &\leq \langle Q_h^\star(x, \cdot) - Q_h^t(x, \cdot), \pi_h^\star(\cdot|x)\rangle_\mathcal{A} \\
&= \mathbb{J}_{\pi_h^\star}(Q_h^\star - Q_h^t)(x).
\end{aligned}
$$

From the Bellman optimality equation and the definition of the temporal-difference, we get

$$
\begin{aligned}
Q_h^\star - Q_h^t &= (r_h + D_h^\rho V_{h+1}^\star) - (r_h + D_h^\rho V_{h+1}^t - \delta_h^t) \\
&= D_h^\rho V_{h+1}^\star - D_h^\rho V_{h+1}^t + \delta_h^t \\
&\leq -D_h^\rho\big(-(V_{h+1}^\star - V_{h+1}^t)\big) + \delta_h^t,
\end{aligned}
$$

where the last inequality follows from Lemma 3. Substituting this in the previous derivation gives us

$$
\begin{aligned}
V_h^\star(x) - V_h^t(x) &\leq \mathbb{J}_{\pi_h^\star}\Big(-D_h^\rho\big(-(V_{h+1}^\star - V_{h+1}^t)\big) + \delta_h^t\Big)(x) \\
&= -\mathbb{J}_{\pi_h^\star} D_h^\rho\big(-(V_{h+1}^\star - V_{h+1}^t)\big)(x) - \mathbb{J}_{\pi_h^\star}(-\delta_h^t)(x). \tag{18}
\end{aligned}
$$

Eq. (18) represents a recursive relation between $V_h^\star - V_h^t$ and $V_{h+1}^\star - V_{h+1}^t$. Then, recursively applying Eq. (18) for all $h \in [H]$ gives

$$
\begin{aligned}
V_1^\star - V_1^t &\leq -\mathbb{J}_{\pi_1^\star} D_1^\rho\big(-(V_2^\star - V_2^t)\big) - \mathbb{J}_{\pi_1^\star}(-\delta_1^t) \\
&\leq -\mathbb{J}_{\pi_1^\star} D_1^\rho\big(-\big(-\mathbb{J}_{\pi_2^\star} D_2^\rho\big(-(V_3^\star - V_3^t)\big) - \mathbb{J}_{\pi_2^\star}(-\delta_2^t)\big)\big) - \mathbb{J}_{\pi_1^\star}(-\delta_1^t) \\
&= -\mathbb{J}_{\pi_1^\star} D_1^\rho\big(\mathbb{J}_{\pi_2^\star} D_2^\rho\big(-(V_3^\star - V_3^t)\big) + \mathbb{J}_{\pi_2^\star}(-\delta_2^t)\big) - \mathbb{J}_{\pi_1^\star}(-\delta_1^t) \\
&= -\Big(\prod_{h=1}^{2} \mathbb{J}_{\pi_h^\star} D_h^\rho\Big)\big(-(V_3^\star - V_3^t)\big) - \sum_{h=1}^{2}\Big(\prod_{i=1}^{h-1} \mathbb{J}_{\pi_i^\star} D_i^\rho\Big)\mathbb{J}_{\pi_h^\star}(-\delta_h^t) \\
&\vdots \\
&\leq -\Big(\prod_{h=1}^{H} \mathbb{J}_{\pi_h^\star} D_h^\rho\Big)\big(-(V_{H+1}^\star - V_{H+1}^t)\big) - \sum_{h=1}^{H}\Big(\prod_{i=1}^{h-1} \mathbb{J}_{\pi_i^\star} D_i^\rho\Big)\mathbb{J}_{\pi_h^\star}(-\delta_h^t) \\
&= -\sum_{h=1}^{H}\Big(\prod_{i=1}^{h-1} \mathbb{J}_{\pi_i^\star} D_i^\rho\Big)\mathbb{J}_{\pi_h^\star}(-\delta_h^t), \tag{19}
\end{aligned}
$$

where the last equality follows due to the fact that $V_{H+1}^\star(x) = V_{H+1}^t(x) = 0$ for all $x \in \mathcal{S}$.

**Term (B).** By definitions of $\delta_h^t, \zeta_{t,h}^1$ and $\zeta_{t,h}^2$ defined in Eq. (16) and Eq. (17), we have

$$
\begin{aligned}
V_h^t(x_h^t) - V_h^{\pi^t}(x_h^t) &= V_h^t(x_h^t) - V_h^{\pi^t}(x_h^t) + \delta_h^t(x_h^t, a_h^t) - \delta_h^t(x_h^t, a_h^t) \\
&= V_h^t(x_h^t) - V_h^{\pi^t}(x_h^t) + D_h^\rho V_{h+1}^t(x_h^t, a_h^t) - D_h^\rho V_h^{\pi^t}(x_h^t, a_h^t)
\end{aligned}
$$

$$+ (Q_h^{\pi^t} - Q_h^t)(x_h^t, a_h^t) - \delta_h^t(x_h^t, a_h^t)$$

$$= (V_h^t - V_h^{\pi^t})(x_h^t) - (Q_h^t - Q_h^{\pi^t})(x_h^t, a_h^t) + D_h^\rho V_{h+1}^t(x_h^t, a_h^t) -$$
$$D_h^\rho V_h^{\pi^t}(x_h^t, a_h^t) - (V_{h+1}^t - V_{h+1}^{\pi^t})(x_{h+1}^t) +$$
$$(V_{h+1}^t - V_{h+1}^{\pi^t})(x_{h+1}^t) - \delta_h^t(x_h^t, a_h^t)$$

$$= (V_{h+1}^t - V_{h+1}^{\pi^t})(x_{h+1}^t) + \xi_{t,h}^1 + \xi_{t,h}^2 - \delta_h^t(x_h^t, a_h^t).$$

Recursively applying the above gives:

$$V_1^t(x_1^t) - V_1^{\pi^t}(x_1^t) = (V_2^t - V_2^{\pi^t})(x_2^t) + \xi_{t,1}^1 + \xi_{t,1}^2 - \delta_1^t(x_1^t, a_1^t)$$

$$= \left( (V_3^t - V_3^{\pi^t})(x_3^t) + \xi_{t,2}^1 + \xi_{t,2}^2 - \delta_2^t(x_2^t, a_2^t) \right) + \xi_{t,1}^1 + \xi_{t,1}^2 - \delta_1^t(x_1^t, a_1^t)$$

$$= (V_3^t - V_3^{\pi^t})(x_3^t) + \sum_{h=1}^2 (\xi_{t,h}^1 + \xi_{t,h}^2) - \sum_{h=1}^2 \delta_h^t(x_h^t, a_h^t)$$

$$\vdots$$

$$= (V_{H+1}^t - V_{H+1}^{\pi^t})(x_{H+1}^t) + \sum_{h=1}^H (\xi_{t,h}^1 + \xi_{t,h}^2) - \sum_{h=1}^H \delta_h^t(x_h^t, a_h^t)$$

$$= \sum_{h=1}^H (\xi_{t,h}^1 + \xi_{t,h}^2) - \sum_{h=1}^H \delta_h^t(x_h^t, a_h^t), \tag{20}$$

where the last equality follows due to the fact that $V_{H+1}^t(x_{H+1}) = V_{H+1}^{\pi^t}(x_{H+1}) = 0$.

Combining Eq. (19) and Eq. (20) gives

$$\mathfrak{R}_T = \sum_{t=1}^T [V_1^\star(x_1^t) - V_1^{\pi^t}(x_1^t)]$$

$$\leq - \sum_{h=1}^H \left( \prod_{i=1}^{h-1} \mathbb{J}_{\pi_i^\star} D_i^\rho \right) \mathbb{J}_{\pi_h^\star}(-\delta_h^t) + \sum_{h=1}^H (\xi_{t,h}^1 + \xi_{t,h}^2) - \sum_{h=1}^H \delta_h^t(x_h^t, a_h^t),$$

which concludes the proof of this lemma.

### D.3 PROOF OF LEMMA 5

Let $\phi : \mathcal{Z} \to \mathcal{H}$ denote the feature representation induced by the kernel $k$, i.e., $k(z, z') = \langle \phi(z), \phi(z') \rangle_{\mathcal{H}}$. For ease of representation, we view $\phi(z)$ as a vector and write $\phi(z)^\top \phi(z') = \langle \phi(z), \phi(z') \rangle_{\mathcal{H}}$ to denote the inner product. The kernel regression problem in Eq. (9) becomes

$$\widehat{\theta} \leftarrow \min_{\theta \in \mathcal{H}} L(\theta) = \sum_{\tau=1}^{t-1} \left[ r_h(x_h^\tau, a_h^\tau) + \hat{\rho}(V_{h+1}^t(\{x'_{(i)}\}_{i=1}^m)) - \theta^\top \phi(x_h^\tau, a_h^\tau) \right]^2 + \lambda \cdot \|\theta\|_{\mathcal{H}}^2. \tag{21}$$

We define the feature matrix $\Phi_h^t : \mathcal{H} \to \mathbb{R}^{t-1}$ and the covariance matrix $\Lambda_h^t : \mathcal{H} \to \mathcal{H}$ as

$$\Phi_h^t = \left[ \phi(z_h^1)^\top, \ldots, \phi(z_h^{t-1})^\top \right]^\top \text{ and } \Lambda_h^t = \sum_{\tau=1}^{t-1} \phi(z_h^\tau) \phi(z_h^\tau)^\top + \lambda I_{\mathcal{H}} = (\Phi_h^t)^\top \Phi_h^t + \lambda I_{\mathcal{H}},$$

where $I_{\mathcal{H}}$ is the identity mapping on $\mathcal{H}$. The Gram matrix $K_h^t$ in Eq. (6) can be expressed as $K_h^t = \Phi_h^t (\Phi_h^t)^\top$, and $k_h^t(z) = \Phi \phi(z)$. With these definitions, we can rewrite Eq. (21) as

$$\min_{\theta \in \mathcal{H}} L(\theta) = \|y_h^t - \Phi_h^t \theta\|_2^2 + \lambda \theta^\top \theta.$$

The solution to the optimization problem above is given by $\widehat{\theta}_h^t = (\Lambda_h^t)^{-1} (\Phi_h^t)^\top y_h^t$. As a result, $\widehat{Q}_h^t$ in Eq. (9) can be expressed as $\widehat{Q}_h^t(z) = \phi(z)^\top \widehat{\theta}_h^t$. In the rest of this section, to further simplify the

notation, we denote $\Phi_h^t$ as simply $\Phi$ when the context is clear. Since $(\Phi\Phi^\top + \lambda I)$ and $(\Phi^\top \Phi + \lambda I_\mathcal{H})$ are positive definite, and thus invertible, and $\Phi^\top(\Phi\Phi^\top + \lambda I) = (\Phi^\top\Phi + \lambda I_\mathcal{H})\Phi^\top$, we have

$$(\Phi_h^t)^{-1}\Phi^\top = (\Phi^\top\Phi + \lambda I_\mathcal{H})^{-1}\Phi^\top = \Phi^\top(\Phi\Phi^\top + \lambda I)^{-1} = \Phi^\top(K_h^t + \lambda I)^{-1}.$$

Consequently, we can write $\widehat{\theta}_h^t$ as

$$\widehat{\theta}_h^t = (\Lambda_h^t)^{-1}\Phi^\top y_h^t = \Phi^\top(K_h^t + \lambda I)^{-1}y_h^t.$$

In the sequel, we will bound the temporal-difference error $\delta_h^t$ defined in Eq. (16). Since $V_h^t(x) = \max_a Q_h^t(x, a)$, we have

$$\delta_h^t = r_h + D_h^\rho V_{h+1}^t - Q_h^t = \mathbb{T}_h^\star Q_{h+1}^t - Q_h^t,$$

where $\mathbb{T}_h^\star$ is the Bellman optimality operator. By Assumption Assumption 1, since for any $t, h$, $Q_{h+1}^t \in [0, H]$, therefore we have $\mathbb{T}_h^\star Q_{h+1}^t \in \mathcal{B}(RH)$. Consequently, there exists $\bar{\theta}_h^t \in \mathcal{H}$ such that for any $z \in \mathcal{Z}$, $\mathbb{T}_h^\star Q_{h+1}^t(z) = \phi(z)^\top\bar{\theta}_h^t$. We can write $\phi(z)$ as

$$\begin{aligned}
\phi(z) &= (\Lambda_h^t)^{-1}\Lambda_h^t\phi(z) \\
&= (\Lambda_h^t)^{-1}(\Phi^\top\Phi + \lambda I_\mathcal{H})\phi(z) \\
&= (\Lambda_h^t)^{-1}(\Phi^\top\Phi)\phi(z) + \lambda(\Lambda_h^t)^{-1}\phi(z) \\
&= \Phi^\top(K_h^t + \lambda I)^{-1}k_h^t(z) + \lambda(\Lambda_h^t)^{-1}\phi(z).
\end{aligned}$$

Using the above, we can write $\phi(z)^\top\bar{\theta}_h^t$ as

$$\phi(z)^\top\bar{\theta}_h^t = k_h^t(z)^\top(K_h^t + \lambda I)^{-1}\Phi\bar{\theta}_h^t + \lambda\phi(z)^\top(\Lambda_h^t)^{-1}\bar{\theta}_h^t.$$

We have:

$$\begin{aligned}
&\phi(z)^\top\widehat{\theta}_h^t - \phi(z)^\top\bar{\theta}_h^t \qquad\qquad\qquad\qquad\qquad\qquad\qquad\qquad (22) \\
&= \phi(z)^\top\Phi^\top(K_h^t + \lambda I)^{-1}y_h^t - k_h^t(z)^\top(K_h^t + \lambda I)^{-1}\Phi\bar{\theta}_h^t - \lambda\phi(z)^\top(\Lambda_h^t)^{-1}\bar{\theta}_h^t \\
&= \underbrace{k_h^t(z)^\top(K_h^t + \lambda I)^{-1}(y_h^t - \Phi\bar{\theta}_h^t)}_{\text{(i)}} - \underbrace{\lambda\phi(z)^\top(\Lambda_h^t)^{-1}\bar{\theta}_h^t}_{\text{(ii)}}.
\end{aligned}$$

We proceed by bounding Term (i) and Term (ii) separately. For Term (ii), by Cauchy-Schwarz inequality:

$$\begin{aligned}
|\text{Term (ii)}| &= |\lambda\phi(z)^\top(\Lambda_h^t)^{-1}\bar{\theta}_h^t| \\
&\leq \|\lambda(\Lambda_h^t)^{-1}\phi(z)\|_\mathcal{H} \cdot \|\bar{\theta}_h^t\|_\mathcal{H} \leq RH\|\lambda(\Lambda_h^t)^{-1}\phi(z)\|_\mathcal{H} \\
&= RH\sqrt{\lambda\phi(z)^\top(\Lambda_h^t)^{-1} \cdot \lambda I_\mathcal{H} \cdot (\Lambda_h^t)^{-1}\phi(z)} \\
&\leq RH\sqrt{\lambda\phi(z)^\top(\Lambda_h^t)^{-1} \cdot (\Lambda_h^t) \cdot (\Lambda_h^t)^{-1}\phi(z)} \\
&= \sqrt{\lambda}RH \cdot b_h^t(z), \qquad\qquad\qquad\qquad\qquad\qquad\qquad\qquad (23)
\end{aligned}$$

where the second last inequality follows from the fact that $(\Lambda_h^t - \lambda I_\mathcal{H})$ is a positive-semidefinite operator, which implies for any $f \in \mathcal{H}$, we have $f^\top(\Lambda_h^t - \lambda I_\mathcal{H})f \geq 0$.

We continue to bound Term (i) in the rest of this section. For $\tau \in [0, t-1]$, the $\tau$-entry of the vector $(y_h^t - \Phi\bar{\theta}_h^t)$ can be expressed as

$$\begin{aligned}
[y_h^t]_\tau - [\Phi\bar{\theta}_h^t]_\tau &= r_h(x_h^\tau, a_h^\tau) + \hat{\rho}(\{V_{h+1}^t(x'_{(i)})\}_{i=1}^m) - \phi(x_h^\tau, a_h^\tau)^\top\bar{\theta}_h^t \\
&= r_h(x_h^\tau, a_h^\tau) + \hat{\rho}(\{V_{h+1}^t(x'_{(i)})\}_{i=1}^m) - (\mathbb{T}_h^\star Q_{h+1}^t)(x_h^\tau, a_h^\tau) \\
&= \hat{\rho}(\{V_{h+1}^t(x'_{(i)})\}_{i=1}^m) - (D_h^\rho V_{h+1}^t)(x_h^\tau, a_h^\tau).
\end{aligned}$$

Combining these above results, we have

$$\begin{aligned}
|\text{Term (i)}| &= |k_h^t(z)^\top(K_h^t + \lambda I)^{-1}(y_h^t - \Phi\bar{\theta}_h^t)| \\
&= |\phi(z)^\top\Phi^\top(K_h^t + \lambda I)^{-1}(y_h^t - \Phi\bar{\theta}_h^t)|
\end{aligned}$$

$$
\begin{aligned}
&= |\phi(z)^\top (\Lambda_h^t)^{-1} \Phi^\top (y_h^t - \Phi \bar{\theta}_h^t)| \\
&= \left| \phi(z)^\top (\Lambda_h^t)^{-1} \left\{ \sum_{\tau=1}^{t-1} \phi(x_h^\tau, a_h^\tau) \cdot \left[ \hat{\rho}(\{V_{h+1}^t(x_{(i)}')\}_{i=1}^m) - (D_h^\rho V_{h+1}^t)(x_h^\tau, a_h^\tau) \right] \right\} \right| \\
&\le \|\phi(z)\|_{(\Lambda_h^t)^{-1}} \cdot \left\| \sum_{\tau=1}^{t-1} \phi(x_h^\tau, a_h^\tau) \cdot \left[ \hat{\rho}(\{V_{h+1}^t(x_{(i)}')\}_{i=1}^m) - \right. \right. \\
&\qquad\qquad\qquad\qquad\qquad\qquad \left. \left. (D_h^\rho V_{h+1}^t)(x_h^\tau, a_h^\tau) \right] \right\|_{(\Lambda_h^t)^{-1}},
\end{aligned}
\tag{24}
$$

where the last inequality follows from the Cauchy-Schwarz inequality. To bound the RKHS norm in the second term, we apply techniques similar to (Yang et al., 2020) by combining concentration of self-normalized processes and uniform convergence over the function classes that contain $V_{h+1}^t$. To achieve this, let us first define the action-value function classes $\mathcal{Q}_{\mathrm{ucb}}(h, R, B)$ as

$$
\begin{aligned}
\mathcal{Q}_{\mathrm{ucb}}(h, R, B) = \{ Q : \\
Q(z) = \min\{ Q_0(z) + \beta \cdot \lambda^{1/2} [k(z,z) - k_{\mathcal{D}}(z)^\top (K_{\mathcal{D}} + \lambda I)^{-1} k_{\mathcal{D}}(z)]^{1/2}, H - h + 1 \}^+, \\
\|Q_0\|_{\mathcal{H}} \le R, \beta \in [0, B], |\mathcal{D}| \le T \}.
\end{aligned}
$$

Let us further define the state-value function classes $\mathcal{V}_{\mathrm{ucb}}(h, R, B)$ as

$$
\mathcal{V}_{\mathrm{ucb}}(h, R, B) = \{ V : V(x) = \max_{a \in \mathcal{A}} Q(x, a) \text{ for } Q \in \mathcal{Q}_{\mathrm{ucb}}(h, R, B) \}.
$$

By (Yang et al., 2020, Lemma C.1), if we set $R_T = 2H\Gamma_k(T, \lambda)$, then we have for all $t \in [T], h \in [H]$, $V_h^t$ as defined in Eq. (9) satisfies that $V_h^t \in \mathcal{V}_{\mathrm{ucb}}(h, R_T, B_T)$ where $B_T$ is defined in Theorem 1.

We now bound Term (i) by a covering number argument over the function classes $\mathcal{V}_{\mathrm{ucb}}(h, R_T, B_T)$ for $h \in [H]$. For any two state-value functions $V, V' : \mathcal{S} \to \mathbb{R}$, we consider the maximum metric (also known as the Chebyshev metric) $d(V, V') = \sup_{x \in \mathcal{S}} |V(x) - V'(x)|$. For $\epsilon, B > 0$, let $N_d(\epsilon, h, B)$ be the $\epsilon$-covering number of $\mathcal{V}_{\mathrm{ucb}}(h, R_T, B)$ with respect to the metric $d$, and $N_\infty(\epsilon, h, B)$ as the $\epsilon$-covering number of $\mathcal{Q}_{\mathrm{ucb}}(h, R_T, B)$ with respect to the maximum metric. Applying (Yang et al., 2020, Lemma E.2) with $\delta = (2T^2 H^3)^{-1}$ and taking a union bound over $h \in [H]$ gives

$$
\begin{aligned}
&\left\| \sum_{\tau=1}^{t-1} \phi(x_h^\tau, a_h^\tau) \cdot \left[ \hat{\rho}(\{V_{h+1}^t(x_{(i)}')\}_{i=1}^m) - (D_h^\rho V_{h+1}^t)(x_h^\tau, a_h^\tau) \right] \right\|_{(\Lambda_h^t)^{-1}}^2 \\
&\le \sup_{V \in \mathcal{V}_{\mathrm{ucb}}(h+1, R_T, B_T)} \left\| \sum_{\tau=1}^{t-1} \phi(x_h^\tau, a_h^\tau) \cdot \left[ \hat{\rho}(\{V(x_{(i)}')\}_{i=1}^m) - (D_h^\rho V)(x_h^\tau, a_h^\tau) \right] \right\|_{(\Lambda_h^t)^{-1}}^2 \\
&\le 2H^2 \log \det(I + K_h^t / \lambda) + 2H^2 t(\lambda - 1) \\
&\quad + 4H^2 \left[ \log N_\infty(\epsilon, h+1, B_T) + \log(2T^2 H^3) \right] + 8t^2 \epsilon^2 / \lambda,
\end{aligned}
$$

uniformly over all $t \in [T], h \in [H]$ with probability $1 - (2T^2 H^2)^{-1}$. The first inequality is due to the fact that $V_{h+1}^t \in \mathcal{V}_{\mathrm{ucb}}(h+1, R_T, B_T)$. According to the algorithm, we set $\lambda = 1 + 1/T$. We further set $\epsilon^* = H/T$, the above simplifies to

$$
\begin{aligned}
&\left\| \sum_{\tau=1}^{t-1} \phi(x_h^\tau, a_h^\tau) \cdot \left[ \hat{\rho}(\{V_{h+1}^t(x_{(i)}')\}_{i=1}^m) - (D_h^\rho V_{h+1}^t)(x_h^\tau, a_h^\tau) \right] \right\|_{(\Lambda_h^t)^{-1}}^2 \\
&\le 4H^2 \Gamma_k(T, \lambda) + 10H^2 + 4H^2 \log N_\infty(\epsilon^*, h+1, B_T) + 12H^2 \log(TH).
\end{aligned}
$$

Combining the above result with Eq. (22), Eq. (23), and Eq. (24), gives

$$
\begin{aligned}
&|\phi(z)^\top (\widehat{\theta}_h^t - \bar{\theta}_h^t)| \\
&\le H b_h^t(z) \cdot \left\{ [4\Gamma_k(T, \lambda) + 10 + 4\log N_\infty(\epsilon^*, h+1, B) + 8\log(TH)]^{1/2} + \sqrt{\lambda} R \right\} \\
&\le H b_h^t(z) \left[ 8\Gamma_k(T, \lambda) + 20 + 8\log N_\infty(\epsilon^*, h+1, B) + 16\log(TH) + 2\lambda R^2 \right]^{1/2} \\
&\le B_T \cdot b_h^t(z) = \beta \cdot b_h^t(z),
\end{aligned}
\tag{25}
$$

holds uniformly for all $t \in [T], h \in [H]$ with probability $1 - (2T^2H^2)^{-1}$. The second inequality follows from the fact that $\sqrt{a} + \sqrt{b} \leq \sqrt{2(a^2 + b^2)}$, and the last inequality follows from the assumption on $B_T$.

Finally, by the definition of the temporal difference error $\delta_h^t$ and Eq. (25), we have:

$$-\delta_h^t(z) = Q_h^t(z) - \mathbb{T}_h^\star Q_{h+1}^t \leq \phi(z)^\top (\widehat{\theta}_h^t - \overline{\theta}_h^t) + \beta b_h^t(z) \leq 2\beta b_h^t(z)$$

which proves the left inequality of Lemma 5. For the right inequality, note that since $Q_{h+1}^t(z) \leq H - h$ for all $z \in \mathcal{Z}$, we have $(\mathbb{T}_h^\star Q_{h+1}^t) \leq H - h + 1$. Therefore, we have

$$\begin{aligned}
\delta_h^t(z) &= \mathbb{T}_h^\star Q_{h+1}^t - Q_h^t(z) \\
&= \phi(z)^\top \overline{\theta}_h^t - \min\{\phi(z)^\top \widehat{\theta}_h^t + \beta \cdot b_h^t(z), H - h + 1\}^+ \\
&\leq \max\{\phi(z)^\top \overline{\theta}_h^t - \phi(z)^\top \widehat{\theta}_h^t - \beta \cdot b_h^t(z), \phi(z)^\top \overline{\theta}_h^t - (H - h + 1)\} \\
&\leq 0,
\end{aligned}$$

where the first term is negative due to Eq. (25), and the second term is negative due to the fact that $(\mathbb{T}_h^\star Q_{h+1}^t) \leq H - h + 1$. This completes the proof of Lemma 5.

### D.4   PROOF OF LEMMA 6

This section follows Yang et al. (2020) to show $\{\zeta_{t,h}^1, \zeta_{t,h}^2\}_{(t,h) \in [T] \times [H]}$ is a bounded martingale difference sequence. We construct the filtration as follows. For any $t \in [T], h \in [H]$, we define the following $\sigma$-algebras

$$\begin{aligned}
\mathcal{F}_{t,h,1} &= \sigma\big(\{(x_i^\tau, a_i^\tau)\}_{(\tau,i) \in [t-1] \times [H]} \cup \{(x_i^t, a_i^t)\}_{i \in [h]}\big), \\
\mathcal{F}_{t,h,2} &= \sigma\big(\{(x_i^\tau, a_i^\tau)\}_{(\tau,i) \in [t-1] \times [H]} \cup \{(x_i^t, a_i^t)\}_{i \in [h]} \cup \{x_{h+1}^t\}\big),
\end{aligned}$$

where $\sigma(\cdot)$ denotes the $\sigma$-algebra generated by a finite set. Since $V_h^t$ and $Q_h^t$ are computed based on trajectories of the first $t - 1$ episodes, they are measurable with respect to $\mathcal{F}_{t,1,1}$. Since the action $a_h^t$ are sampled from $\pi_t(\cdot | x_h^t)$, we have

$$\mathbb{E}\big[\zeta_{t,h}^1 | \mathcal{F}_{t,h-1,2}\big] = 0.$$

Thus, applying Azuma-Hoeffding inequality (Azuma, 1967), we obtain that for all $t > 0$:

$$\mathbb{P}\Big(\big|\sum_{t=1}^T \sum_{h=1}^H \zeta_{t,h}^1\big| > t\Big) \leq 2\exp\Big(\frac{-t^2}{16TH^3}\Big). \tag{26}$$

We let the right hand side equal to $\delta/2$, yielding $t = \sqrt{16TH^3 \cdot \log(4/\delta)}$.

Next, we bound $\zeta_{t,h}^2$ using the risk estimator concentration inequality. Recall that the empirical risk estimate $\hat{\rho}$ achieves the rate of $\Xi(m, \delta)$, i.e.,

$$\mathbb{P}\big(|\rho(Z) - \hat{\rho}(\{Z_i\}_{i=1}^m)| \leq \Xi(m, \delta)\big) \geq 1 - \delta.$$

By definition, $D_h^\rho V_{h+1}^t(x_h^t, a_h^t) = \rho(x')$ where $x' \sim \mathbb{P}_h(\cdot | x_h^t, a_h^t)$. Note that $x_{h+1}^t$ is also sampled from $\mathbb{P}_h(\cdot | x_h^t, a_h^t)$. Applying the above inequality, for all $t \in [T], h \in [H]$, we have

$$\mathbb{P}\Big(\big|D_h^\rho V_{h+1}^t(x_h^t, a_h^t) - V_{h+1}^t(x_{h+1}^t)\big| \leq \Xi(m, \delta/(4TH))\Big) \geq 1 - \delta/(4TH). \tag{27}$$

Taking a union bound over all $t \in [T]$ and $h \in [H]$ yields

$$\mathbb{P}\Big(\sum_{t=1}^T \sum_{h=1}^H \big|D_h^\rho V_{h+1}^t(x_h^t, a_h^t) - V_{h+1}^t(x_{h+1}^t)\big| \leq TH \cdot \Xi(m, \delta/(4TH))\Big) \geq 1 - \delta/4. \tag{28}$$

Following similar arguments, we have

$$\mathbb{P}\Big(\sum_{t=1}^T \sum_{h=1}^H \big|D_h^\rho V_{h+1}^{\pi_t}(x_h^t, a_h^t) - V_{h+1}^{\pi_t}(x_{h+1}^t)\big| \leq TH \cdot \Xi(m, \delta/(4TH))\Big) \geq 1 - \delta/4. \tag{29}$$

Finally, performing a union bound over 26, 28, and 29 gives us with probability at least $1 - \delta$, we have

$$\sum_{t=1}^T \sum_{h=1}^H (\zeta_{t,h}^1 + \zeta_{t,h}^2) \leq \sqrt{16TH^3 \log(2/\delta)} + 2TH \cdot \Xi(m, \delta/(4TH)),$$

which concludes the proof of Lemma 6.

### D.5 PROOF OF COROLLARY 1

We first restate CVaR empirical risk estimator and its concentration result.

**Lemma 7.** (Lemma 3 in Yu et al. (2018)) *Let $Z_1, \ldots, Z_m \sim F_Z$ be $m$ i.i.d. bounded random variables, i.e., $\mathbb{P}[0 \leq Z_i \leq B] = 1, \forall i$, then we have*

$$\mathbb{P}\Big[\big|\text{CVaR}_\alpha(Z) - \widehat{\text{CVaR}}_\alpha(\{Z_i\}_{i=1}^m)\big| \geq \varepsilon\Big] \leq 2\left(1 + \frac{4}{\varepsilon(1-\alpha)}\right) \exp\left[\frac{-m\varepsilon^2(1-\alpha)^2}{2(2-\alpha)^2 B^2}\right],$$

*where*

$$\widehat{\text{CVaR}}_\alpha(\{Z_i\}_{i=1}^m) = \max_{s \in \mathbb{R}}\left\{s + \frac{1}{\alpha m}\sum_{i=1}^m (Z_i - s)^-\right\}.$$

Setting the RHS to $\delta$, we have

$$\delta = 2\left(1 + \frac{4}{\varepsilon(1-\alpha)}\right) \exp\left[\frac{-m\varepsilon^2(1-\alpha)^2}{2(2-\alpha)^2 B^2}\right],$$

or equivalently,

$$m = \log\left(\frac{4TH(\epsilon(1-\alpha)+4)}{\delta\epsilon(1-\alpha)} \frac{2H^2(2-\alpha)^2}{\epsilon^2(1-\alpha)^2}\right).$$

For the regret to be order-optimal, we need

$$\epsilon = O\left(\frac{B_T\sqrt{\Gamma_k(T,\lambda)}}{\sqrt{T}}\right) \text{ and } \delta = (8T^3H^3)^{-1}.$$

That gives us

$$m = O\left(\log\left(\frac{T^5 H^6}{B_T^2 \Gamma_k(T,\lambda)}\right)\right),$$

which concludes the proof of Corollary 1.

## E ADDITIONAL EXPERIMENTS

This section provides additional empirical results to demonstrate the effectiveness of RA-UCB. We study the proposed algorithm under various risk measures, namely VaR, CVaR, and EVaR. Refer to Appendix C for a formal introduction and discussion of these risk measures.

The robot navigation environment is similar to the setting in Section 5.1. The robot receives a positive reward of 10 for reaching the destination and a negative reward for being close to obstacles. The negative reward increases exponentially as the robot comes close to the obstacles. We set the horizon of each episode to $H = 30$ and use $m = 100$ samples from the weak simulator to estimate the risk in Eq. (7). We approximate the state-action value function using the RBF kernel and the `KernelRidge` regressor from Scikit-learn.

We run RA-UCB for 50 episodes and report the performance of the learned policy with three different risk measures, namely Value-at-Risk (VaR), Conditional Value-at-Risk (CVaR), and Entropic Value-at-Risk (EVaR). Each risk measure is evaluated against different values of the risk parameter $\alpha \in [0.1, 0.5, 0.9]$. We note that VaR is not a coherent risk measure; therefore, our regret upper bound does not apply to VaR. Regardless, VaR is still an important risk measure with many real-world applications and has important connections with CVaR and EVaR.

Fig. 4 shows the robot's cumulative rewards following the learned policies after 50 episodes. Each column represents a different risk measure, and each row represents a different risk parameter. We observe that the reward distribution changes when we vary the risk parameters. For $\alpha = 0.1$ (top row), the policies are more risk-averse, favoring safer paths with higher worst-case rewards but having smaller average rewards. As we increase the value of $\alpha$, $\alpha = 0.5$ (middle row), and $\alpha = 0.9$ (bottom row), the learned policies become more risk-tolerant, which causes the average rewards to be higher but occasionally small.

We also observe that, for the same risk parameter, CVaR outperforms VaR marginally in terms of average rewards. This gain is because VaR does not control rewards below its value (for instance, you

can significantly reduce the smallest rewards below VaR, but the Value-at-Risk will not change). In other words, VaR disregards some parts of the distribution. This property can be both good and bad, depending upon the applications. For example, VaR estimates are statistically more stable than CVaR estimates. However, in our case, we use a sufficiently high number of samples to estimate the risk values; thus, risk measures that consider the complete distribution, like CVaR or EVaR, are slightly more effective.

Furthermore, it can be shown that for the same risk parameter $\alpha$, EVaR is more risk-averse than both VaR and CVaR (Ahmadi-Javid, 2012). Our results show that the reward distribution using EVaR (plots the rightmost columns) tends to have lower average rewards but better worst-case rewards.

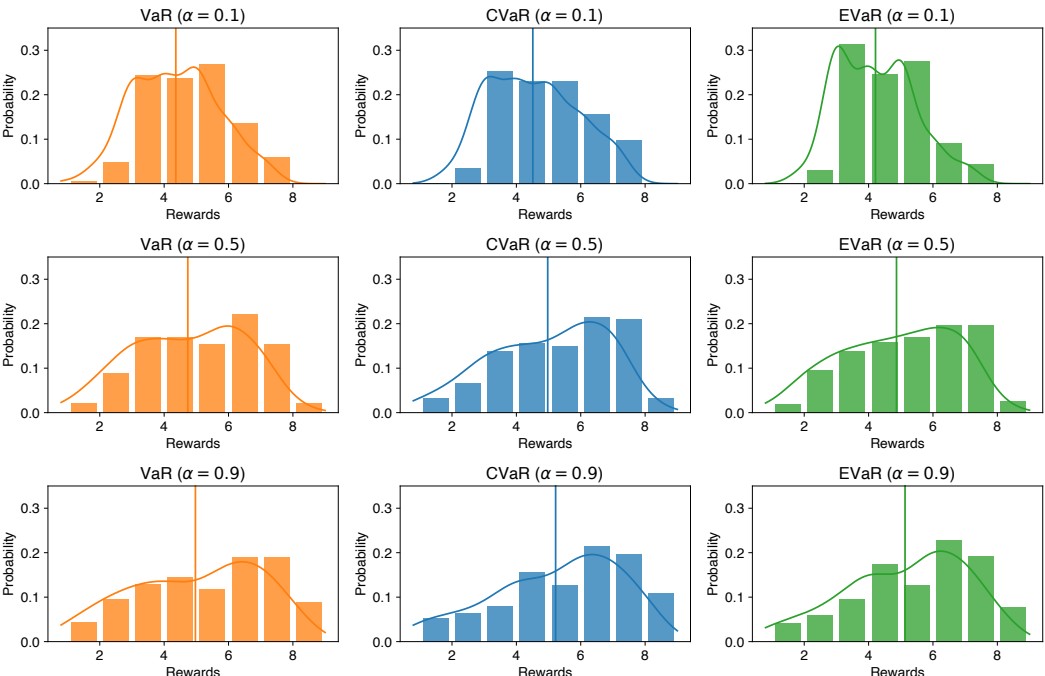

Figure 4: Histograms show the distribution of the cumulative reward when following the learned policy. The solid lines represent the estimated densities using Gaussian kernels, and the vertical lines represent the mean of the distribution. For $\alpha = 0.1$ (top row), the policy is more risk-averse, favoring safer paths with higher worst-case rewards but having smaller average rewards. As we increase $\alpha$, $\alpha = 0.5$ (middle row), and $\alpha = 0.9$ (bottom row), the learned policy becomes more risk-tolerant, which causes the average rewards to be higher but occasionally small. Furthermore, for the same risk parameters, e.g., $\alpha = 0.1$, EVaR usually has the smallest average but better worst-case rewards. This result also confirms that EVaR is more risk-averse than VaR and CVaR for the same risk parameter (Ahmadi-Javid, 2012).

