# OpenReview forum: "Risk-Aware Reinforcement Learning with Coherent Risk Measures and Non-linear Function Approximation"
_ICLR.cc/2023/Conference — ICLR 2023 poster_

### Official Review · Reviewer_osfS · 2022-10-18

**Confidence:** 3
**Correctness:** 4
**Technical Novelty And Significance:** 3
**Empirical Novelty And Significance:** 2
**Recommendation:** 6

**Clarity, Quality, Novelty And Reproducibility:**

- The paper is well-written in general, and the sections are well-organized.
- The authors claim that they first formalize the risk-aware RL setting with coherent risk measures. Does it mean the form of the regret function?
- It is not yet clear to me whether the class of coherent risk measures contains important measures other than CVaR.
- The algorithm RA-UCB seems to be a simple variant of the classic UCB algorithm, which is lack novelty by itself.

**Strength And Weaknesses:**

**Strength:**
- The paper provides a risk-aware RL algorithm under coherent risk measures with a sub-linear performance guarantee.
- The authors overcome the challenge of the non-linear Bellman equations through the application of the super additivity property of the measures.
- A unified framework is proposed for analyzing regrets under coherent risk measures.
- A number of empirical experiments on both synthetic and real-world datasets showcase the performance of the proposed algorithm.

**Weakness:**
- There is no matching lower bound provided to show the optimality of the proposed method, and I tend to believe that the algorithm is not optimal in that sense.
- The algorithm requires an oracle (or simulator) to generate an unbiased estimation of the value functions, and such simulators may not exist in some of the use cases.

**Summary Of The Paper:**

The paper proposed and analyzed the risk-aware reinforcement learning algorithm RA-UCB with coherent risk measures and non-linear function approximation.

**Summary Of The Review:**

Overall, I feel like this is a solid paper that provides a new approach to analyzing risk-aware RL algorithms under a particular class of risk measures that have never been touched before, which adds to the existing literature in the study of risk-sensitive RL.

===============================

The rebuttal from the authors address all my questions. I feel like the paper is in general well-written, but the choice of the objective function, as suggested by reviewer aLWD, is still a bit concerning. Therefore, I decide to keep my original score for the paper.

---

> ### Author Response · Authors · 2022-11-18
> **Response to Reviewer osfS (Part 2)**
>
> > It is not yet clear to me whether the class of coherent risk measures contains important measures other than CVaR.
>
> There are other risk measures like Entropic Value-at-Risk, Tail Value-at-Risk, Proportional Hazard (PH) risk measure, g-Entropic risk measures, Wang risk measure, and Superhedging price belonging to the family of coherent risk measures. These risk measures have important applications. For example, Proportional Hazard (PH) risk measure is widely used in healthcare domains such as clinical trials (Rulli et al., 2018 [1]) or epidemiology (Moolgavkar et al., 2018 [2]). Wang risk measure and Superhedging price are commonly used in financial applications such as asset pricing (Wang, 2000 [3]) or portfolio optimization (Löhne & Rudloff, 2014 [4]).
>
> In the RL context, CVaR is the most well-known and commonly used among all coherent risk measures and is relatively well-studied in the literature (Bäuerle & Ott, 2011; Yu et al., 2018). Some very recent works (Ni & Lai 2022 [5]; 2022 September [6]) have started investigating the use of EVaR in RL. Unlike our work, the techniques used by Ni & Lai (2022 [5]; 2022 September [6]) exploit properties that are unique to EVaR and thus cannot be generalized to others in the family of coherent risk measures.
>
> We hope that our work can inspire further investigation of other risk measures in the context of RL.
>
> [1] Rulli, E., Ghilotti, F., Biagioli, E., Porcu, L., Marabese, M., D’Incalci, M., ... & Torri, V. (2018). Assessment of proportional hazard assumption in aggregate data: a systematic review on statistical methodology in clinical trials using time-to-event endpoint. British journal of cancer, 119(12), 1456-1463.
>
> [2] Moolgavkar, S. H., Chang, E. T., Watson, H. N., & Lau, E. C. (2018). An assessment of the Cox proportional hazards regression model for epidemiologic studies. Risk Analysis, 38(4), 777-794.
>
> [3] Wang, S. S. (2000). A class of distortion operators for pricing financial and insurance risks. Journal of risk and insurance, 15-36.
>
> [4] Löhne, A., & Rudloff, B. (2014). An algorithm for calculating the set of superhedging portfolios in markets with transaction costs. International Journal of Theoretical and Applied Finance, 17(02), 1450012.
>
> [5] Ni, X., & Lai, L. (2022). EVaR optimization for risk-sensitive reinforcement learning. IEEE Transactions on Information Theory.
>
> [6] Ni, X., & Lai, L. (2022, September). Policy Gradient Based Entropic-VaR Optimization in Risk-Sensitive Reinforcement Learning. In 2022 58th Annual Allerton Conference on Communication, Control, and Computing (Allerton) (pp. 1-6). IEEE.
>
> > The algorithm RA-UCB seems to be a simple variant of the classic UCB algorithm, which is lack novelty by itself.
>
> We want to highlight a novel aspect of RA-UCB compared to the other UCB-based algorithms in standard RL. Our primary novelty in algorithm design lies in constructing the response vector in Eq. (7), which enables the regret analysis in Theorem 1.
>
> **Eq. (7) implements the one-step Bellman optimality update in Eq. (4):**
> To see this, let $X' \sim P_h(\cdot|x^\tau_h,a^\tau_h)$ be the random variable representing the next state. Recall that $V^t_{h+1}$ is the estimated value function by our algorithm at episode $t$. Thus, $V^t_{h+1}(X')$ is also a random variable where the randomness comes from $X'$. Here, we can start looking at $\rho(V^t_{h+1}(X'))$, i.e., the risk measure $\rho$ applied on the random variable $V^t_{h+1}(X')$. Intuitively, one can view this as the **risk-adjusted value of the next state**. The second term in Eq. (7), i.e., $\widehat{\rho}(\\{V^t\_{h+1}(x'\_{(i)})\\}\_{i=1}^m)$, is an empirical estimate of $\rho(V^t\_{h+1}(X'))$.
>
> The response vector in Eq. (7) is constructed such that the regret admits a new decomposition (please refer to Lemma 4 in Appendix D.1) that allows us to control all the resulting terms effectively, consequently leading to the first sub-linear regret algorithm for risk-aware RL with coherent risk measures.

---

> ### Author Response · Authors · 2022-11-18
> **Response to Reviewer osfS (Part 1)**
>
> Thank you for your detailed comments and positive feedback. We have responded to your comments and questions below.
>
> > There is no matching lower bound provided to show the optimality of the proposed method, and I tend to believe that the algorithm is not optimal in that sense.
>
> We agree that a lower bound can tell whether any regret upper bound is optimal or sub-optimal. Hence, deriving a lower bound for the risk-aware RL setting is highly desirable and an active research direction in our agenda. However, unlike the upper bound, we have yet to find a unified analysis for the entire class of coherent risk measures.
>
> > The algorithm requires an oracle (or simulator) to generate an unbiased estimation of the value functions, and such simulators may not exist in some of the use cases.
>
> We agree with the reviewer that alleviating our assumption of the weak simulator is an important research direction. However, the challenge with risk-aware RL comes from the fact that some coherent risk measure like CVaR depends on other higher moments of the distribution, which **cannot be estimated using a single sample**; this is in a similar vein to saying that one cannot obtain an unbiased variance estimate from a single sample.
> The previous works that considered risk-aware RL with CVaR (Bäuerle and Ott, 2011; Yu et al., 2018) in the **tabular** setting even assume that the transition kernel is known in advance. Our weak simulator assumption is the **weakest assumption** made among similar risk-aware RL settings and holds in many real-life applications; for example, after taking action in a given state of a game (chess or go), the set of possible next states is known to the players.
>
> > The authors claim that they first formalize the risk-aware RL setting with coherent risk measures. Does it mean the form of the regret function?
>
> The first bullet point in our contributions refers to our proposal to study the risk-aware RL problem with coherent risk measures in the **regret minimization** framework and our choice of the risk-aware objective function given in Eq. (2).
> Since regret is defined as the sub-optimality of a policy with respect to the optimal policy, it is crucial to first show the existence of this optimal policy to ensure that the notion of regret is properly defined. This need motivates our choice of the nested composition of risk in Eq. (2) which guarantees the existence of the optimal Markov policy (Theorem 4, Ruszczyński (2010)).

---

> ### Author Response · Authors · 2022-11-25
> **A friendly reminder**
>
> Dear Reviewer osfS,
>
> We want to thank you for your encouraging comments. We have addressed your questions in the response and incorporated them in the revised manuscript.
>
> Please let us know if you have any more questions, and we would be happy to address them within our allowed period.
>
> Warmest regards,
> Authors of Paper1497.

---

> ### Author Response · Authors · 2022-11-28
> **More clarification on the choice of the objective function**
>
> Dear Reviewer osfS,
>
> Thank you very much for acknowledging our responses.
>
> > I feel like the paper is in general well-written, but the choice of the objective function, as suggested by reviewer aLWD, is still a bit concerning.
>
> We would like to clarify a bit further on our choice of the objective function in Equation 2. There are two main reasons behind this choice.
>
> 1. **This leads to many favorable theoretical properties.** Firstly, this choice guarantees the existence of an optimal Markov policy, which is critical to study the risk-aware RL problem in the regret minimization framework. This is because regret is defined as the sub-optimality of a policy with respect to the optimal policy, it is crucial to first show the existence of this optimal policy to ensure that the notion of regret is properly defined. Secondly, this choice leads to a *time-consistent* evaluation of the risk measure, which was introduced briefly in the Response to Reviewer aLWD (Part 1), and explained more formally in Appendix B in the manuscript. This property ensures that we do not contradict ourselves in our risk evaluation. If we observe the same realization, the sequence that is better today should continue to be better tomorrow. Our risk preference stays the same over time. Note that this property is trivially satisfied in standard RL where the risk measure is replaced with expectation. In contrast, a single-stage risk measure (i.e., *static* version) applied on the cumulative reward $\rho\big(\sum_{h=1}^{H}r_h(x_h,a_h)\big)$ does not enjoy this *time consistency* property (Ruszczyński, 2010).
>
> 2. **This is widely adopted in the risk-aware RL literature.** The same objective function has been widely used in the risk-aware RL literature by Bäuerle & Ott, 2011; Yu et al., 2018; Rigter et al., 2021.
>
> We hope the above discussion sheds some light on why we use the dynamic risk measure (as defined in Eq. (2)) over the static version. Please let us know if you have further concerns/questions. If not, we sincerely hope that you can consider improving your opinion of our work.
>
> Thank you again.
>
> Warmest regards,
> Authors of Paper1497.

---

### Official Review · Reviewer_yVV1 · 2022-10-23

**Confidence:** 3
**Correctness:** 4
**Technical Novelty And Significance:** 3
**Empirical Novelty And Significance:** 3
**Recommendation:** 8

**Clarity, Quality, Novelty And Reproducibility:**

### Clarity and Quality
The paper is well-structured. The definitions are clear and well-motivated. The proof sketch is easy to follow.

### Novelty
The main results are novel, to the best of the reviewer's knowledge

### Reproducibility

Since the main result is on the theory side, I am not worried about reproducibility. The code has been uploaded, and I think it is fine.

**Strength And Weaknesses:**


### Strength
1. The paper is well-structured. The definitions are clear and well-motivated. The proof sketch is easy to follow.

2. To the best of the reviewer's knowledge, the RA-UCB algorithm is novel. The way it defines means and variances of action values is inspiring, especially after acknowledging their similarity with the Gaussian process regression and linear bandits problem, ** although more explanation about the insights is appreciated.**

3. The framework for analyzing the regret of risk-aware RL policy can be scaled to coherent risk measures, as opposed to Fei et al. (2021) which used the entropic risk measure. The theoretical results are meaningful and important, which leads to the first sub-linear regret upper bound of the risk-aware RL policy with coherent risk measures.


### Weaknesses

1. The algorithm assumes access to a weak simulator. Although this assumption is less influential compared to the ones in previous works, the algorithm becomes no longer model-free and how to alleviate the assumption will be an important direction.

2. Many previous works have explored implementing risk-sensitive RL with non-linear function approximation. While the related works section covers most of them, it is still unclear whether the difference is significant.

3. The empirical results are weak, but I won't argue for more since the main contribution is on the theory side.


**Summary Of The Paper:**

This paper introduces a framework for analyzing the regret of risk-aware RL policy with coherent risk measures. The framework is based on an episodic finite-horizon Markov decision process (MDP) instead of the MDP with a discounted factor and infinite horizon. Under this framework, this paper proposes a Risk-Aware Upper Confidence Bound (RA-UCB) for performing value iteration under a coherent risk measure by applying non-linear function approximation. The theoretical results provide the regret upper bound guarantee of RA-UCB. To develop this bound, this paper decomposes the risk-aware RL policy’s egret by the super-additivity property of coherent risk measures and assumes access to a weak simulator. Some empirical results are presented to empirically demonstrate the performance of this algorithm.

**Summary Of The Review:**

The quality of this paper is satisfying. The main contribution is presented clearly with strong motivations. The algorithm is novel, and the theoretical results are meaningful. The reviewer lists some weaknesses, but they are minor compared to the contributions. After reviewing the main paper and scanning through the appendix, the reviewer suggests accepting this paper, although it is possible that reviewer dismisses some important problems.

---

> ### Author Response · Authors · 2022-11-18
> **Response to Reviewer yVV1**
>
> We thank you for your encouraging comments and suggestions. We have responded to your comments below.
>
> > The algorithm assumes access to a weak simulator. Although this assumption is less influential compared to the ones in previous works, the algorithm becomes no longer model-free and how to alleviate the assumption will be an important direction.
>
> We agree with the reviewer that alleviating the assumption of the weak simulator is an important research direction. However, the challenge with risk-aware RL comes from the fact that some coherent risk measure like CVaR depends on other higher moments of the distribution, which **cannot be estimated using a single sample**; this is in a similar vein to saying that one cannot obtain an unbiased variance estimate from a single sample.
> The previous works that considered risk-aware RL with CVaR (Bäuerle and Ott, 2011; Yu et al., 2018) in the **tabular** setting even assume that the transition kernel is known in advance. Our weak simulator assumption is one of the **weakest assumptions** made among similar risk-aware RL works and holds in many real-life applications; for example, after taking action in a given state of a game (chess or go), the set of possible next states is known to the players.
>
> > Many previous works have explored implementing risk-sensitive RL with non-linear function approximation. While the related works section covers most of them, it is still unclear whether the difference is significant.
>
> Though risk-aware RL with specific coherent risk measures has been studied extensively by both the RL and Operations Research communities, our work provides the **first algorithm with sub-linear regret guarantees** for the entire class of coherent risk measures. Establishing a regret upper bound in the risk-aware setting is met with several challenges. We want to highlight our technical innovation in two major parts of the analysis that are crucial to the proof of Theorem 1:
>
> 1. **Regret decomposition in Lemma 1.** The standard first step to analyze RL regret bound often involves decomposing the regret into a sum of discrepancies terms over each episode $t \in [T]$ and each time step $h \in [H]$. Standard regret decomposition (Lemma 5.1 in Yang et al., 2020) relies heavily on the *linearity of expectation* property of the risk-neutral Bellman equation. This property **does not hold** for the risk-aware RL setting when the expectation in the Bellman equation is replaced with the CVaR risk measure. We overcome this challenge by leveraging the super-additivity of the CVaR risk measure. More details are in Lemma 3 in Appendix D and the paragraph that follows.
>
> 2. **Analysis to control the growth of $\xi^2_{t,h}$ (Term II of Lemma 4, defined in Eq. (17)).** In risk-neutral RL, one can show that $\\{\xi^2\_{t,h}\\}\_{t\in[T], h\in[H]}$ is a bounded martingale difference sequence and apply Azuma-Hoeffding concentration inequality. However, this sequence is no longer a martingale in the risk-aware setting. To overcome this challenge, we **design the RA-UCB algorithm** such that **the regret admits a new decomposition that allows us to control $\xi^2\_{t,h}$** using the concentration inequality of the risk estimator.
> More details are in the proof of Lemma 6 in Appendix D.4.
>
> In summary, our novel technical innovations mentioned above, coupled with the algorithm design of RA-UCB, play a critical role in establishing the regret upper bound in Theorem 1. We believe these novel ideas will be of interest to the community, especially in the theoretical aspects of risk-aware RL.

---

> > ### Comment · Reviewer_yVV1 · 2022-11-26
> > **Reviewer response**
> >
> > Sorry for my late reply. Thanks for addressing the issues. Recently I have spent time covering some prior works mentioned in the paper. Although there are some similar works as mentioned by other reviewers, I think the paper is generally in a good shape. The novel is sufficient in my mind and I am happy to see the acceptance of this paper.

---

> ### Author Response · Authors · 2022-11-25
> **A friendly reminder**
>
> Dear Reviewer yVV1,
>
> We want to thank you for your encouraging comments. We have addressed your questions in the response and incorporated your suggestions in the revised manuscript.
>
> Please let us know if you have any more questions, and we would be happy to address them within our allowed period.
>
> Warmest regards,
> Authors of Paper1497.

---

### Official Review · Reviewer_aLWD · 2022-10-25

**Confidence:** 3
**Correctness:** 4
**Technical Novelty And Significance:** 3
**Empirical Novelty And Significance:** Not applicable
**Recommendation:** 6

**Clarity, Quality, Novelty And Reproducibility:**

Overall I found the technical exposition quite dense and hard to follow.  While there were some references to the prior literature, I also found it hard to distinguish which parts of the analysis are novel.  For example, how much of the analysis of the first term in Theorem 1 is standard?  I found equations (6)-(8) quite hard to parse as limited intuition is given for the roles all the different pieces are playing.

Since the main story of the paper is around estimating the risk-aware update, the biggest issue of this type for me was around how this is actually done and how performance depends on the number of samples.  The latter is wrapped up in a function and presumably buried somewhere in the appendix.  Corollary 1 gives a sufficient number of sample only for one special case.  In the experiments I can’t actually figure out how many samples were used!  The computational complexity part is confusing because it says the memory is O(N^2) where N is the number of samples (which is m earlier).  But then it says this is O(t^2) for episode t and I don’t understand why the number of samples should vary with the episode number.  Looking at (6) I can see how I the memory used by K^t should scale with quadratically t, but the definitions just list an arbitrary number of samples m so I don’t see where these are being scaled.  I don’t even quite understand the details of how (7) actually makes use of the samples; I suspect it would help to unpack some of the definitions for the reader.

In a more minor note, while I have seen the form of regret in (5) used before, my sense is that it is at least somewhat unusual (though perhaps standard in a certain sub-literature and needs some references and discussion.

**Strength And Weaknesses:**

On this positive side I’m appreciative of the overall agenda of risk-aware RL.  I agree that coherent risk measures in particular are interesting.  The idea of using a weak simulator to enable estimating them is nice.  So overall I am quite positive about the theoretical results.

On the negative side, the paper lacks a related work section.  I was left wondering about the relationship to work such as distributional RL.  While there is a related work section in the appendix, for the purposes of this review I am ignoring it as to do otherwise would be abusive of the page limits.  I do not believe this paper is complete without a proper discussion of related work.

I found the clarity of presentation to be a notable negative.  I discuss this in more detail below, but I have a hard time digging out the insights behind the way the weak simulator is used from the presentation.  On a technical level, I am somewhat concerned about the form of the objective given by equation (2).  The justifications provided for it seem to center around technical convenience (e.g. guaranteeing the optimal policy is Markov).  But how does this form of the objective affect the “meaning” of the risk measure chosen relative to the “static” version?  Finally, while a smaller issue for a paper whose contributions are largely theoretical, it is a bit disappointing that after a pitch about how prior work was specialized for CVaR but this approach was more general, all the experiments only used CVaR.


**Summary Of The Paper:**

This paper studies the problem of RL for finite horizon MDPs with possibly infinite state spaces to learn the policy which is optimal according to an objective determined by a coherent risk measure rather than the standard expected return objective.  The proposed algorithm, RA-UCB, learns a RKHS representation for the Q-value function.  A key technical challenge is that a single sample does not suffice to form an estimate of the risk measure, so an assumption is used that a “weak simulator” which can sample the transition dynamics.  While the primary contribution is theoretical, small simulation demonstrate an implementation with CVaR as the risk measure.

**Summary Of The Review:**

Overall I like the question asked and technical contributions of the paper, but I have some small technical questions and substantive concerns with the presentation.

---

> ### Author Response · Authors · 2022-11-18
> **Response to Reviewer aLWD (Part 4)**
>
> > Corollary 1 gives a sufficient number of samples only for one special case. In the experiments, I can’t actually figure out how many samples were used! The computational complexity part is confusing because it says the memory is O(N^2) where N is the number of samples (which is m earlier). But then it says this is O(t^2) for episode t and I don’t understand why the number of samples should vary with the episode number. Looking at (6) I can see how the memory used by K^t should scale with quadratically t, but the definitions just list an arbitrary number of samples m so I don’t see where these are being scaled.
>
> Corollary 1 gives a sufficient number of samples for the **entire** $T$ episodes, whereas $m$ is the number of samples to estimate a single element in the response vector. Therefore, following the theory, we should set $m$ to be roughly
> $$
>     m \asymp \log{\big({T^5 H^6}/{B_T^2 \Gamma_k(T,\lambda)}\big)}\ .
> $$
> In our experiments, we observe that the value of **$m=100$** gives a competitive performance.
>
> We have revised the computational complexity to include the dependence on $m$ (i.e., the number of samples from the weak simulator) in the revised paper. We re-state the updated content here for convenience:
>
> **Computational complexity of RA-UCB:**
> In the $t$-th episode, we need to solve $H$ kernel ridge regression problems. Each regression problem complexity is dominated by two operations:
> 1. Inversion of Gram matrix $K^t_h$ of size $(t-1)\times(t-1)$ in Eq. (6) has $O(t^3)$ time complexity and $O(t^2)$ space complexity.
> 2. Construction of the response vector in Eq. (7) has $O(mt)$ time and space complexities.
>
> Therefore, the time and space complexities of the $t$-episode are $O(H(t^3 + mt))$ and $O(H(t^2 + mt))$, respectively.
>
> > I have seen the form of regret in (5) used before, my sense is that it is at least somewhat unusual (though perhaps standard in a certain sub-literature and needs some references and discussion.
>
> We are using the widely-adopted notion of regret in the statistical RL theory community. Our notion of regret compares the current learned policy's performance (i.e., value of the risk-aware state-value function at the initial state) against that of the optimal policy. The same notion of regret is widely used in the risk-neutral RL setting (Jin et al., 2020; Yang et al., 2020) and risk-aware RL setting (Fei et al., 2021; Fei & Xu, 2022).

---

> ### Author Response · Authors · 2022-11-18
> **Response to Reviewer aLWD (Part 3)**
>
> > I found equations (6)-(8) quite hard to parse as limited intuition is given for the roles all the different pieces are playing. I don’t even quite understand the details of how (7) actually makes use of the samples; I suspect it would help to unpack some of the definitions for the reader. (*Also, the response to the comment:* I have a hard time digging out the insights behind the way the weak simulator is used from the presentation.)
>
> The main novelty in our algorithm design is represented by Eq. (7) which enables the regret analysis in Theorem 1. Therefore, we parse Eq. (7) and explain how the samples from the weak simulator are utilized. For completeness and self-containment, we now re-state Eq. (7) given in the paper. Given the observed histories and the weak simulator, we defined the response vector $y^t_h \in \mathbb{R}^{t-1}$ as
>
> $$[y^t\_h] = [r_h(x^\tau\_h,a^\tau\_h) + \widehat{\rho}(\\{V^t\_{h+1}(x'\_{(i)})\\}\_{i=1}^m)]\_{\tau\in[t-1]}$$
>
> where $\\{x'\_{(i)}\\}_{i=1}^m$ are $m$ next states drawn from the **weak simulator** $P_h(\cdot|x^\tau_h,a^\tau_h)$.
>
> **Eq. (7) implements the one-step Bellman optimality update in Eq. (4):**
> To see this, let $X' \sim P_h(\cdot|x^\tau_h,a^\tau_h)$ be the random variable representing the next state. Recall that $V^t_{h+1}$ is the estimated value function by our algorithm at episode $t$. Thus, $V^t_{h+1}(X')$ is also a random variable where the randomness comes from $X'$. Here, we can start looking at $\rho(V^t_{h+1}(X'))$, i.e., the risk measure $\rho$ applied on the random variable $V^t_{h+1}(X')$. Intuitively, one can view this as the **risk-adjusted value of the next state**. The second term in Eq. (7), i.e., $\widehat{\rho}(\\{V^t\_{h+1}(x'\_{(i)})\\}\_{i=1}^m)$, is an empirical estimate of $\rho(V^t_{h+1}(X'))$.
>
> **Eq. (8) is the solution to the following optimization problem:**
> At the $t$-episode and at time step $h$, let us consider the following kernel ridge regression problem:
>
> $$ \DeclareMathOperator*{\argmin}{arg\\,min} \argmin_{f \in \mathcal{H}} \bigg[ \sum_{\tau=1}^{t-1} \big[ y^t_h - f(x^\tau_h, a^\tau_h) \big]^2 + \lambda \cdot \lVert f \rVert^2_{\mathcal{H}} \bigg]$$
>
> where the target is the response vector $y^t_h$ discussed above and $\lambda > 0$ is a regularization parameter. Then, $\mu^t_h$ defined in Eq. (8) is the solution to the above minimization problem:
>
> $$ \DeclareMathOperator*{\argmin}{arg\\,min} \mu^t_h \in \argmin_{f \in \mathcal{H}} \bigg[ \sum_{\tau=1}^{t-1} \big[ y^t_h - f(x^\tau_h, a^\tau_h) \big]^2 + \lambda \cdot \lVert f \rVert^2_{\mathcal{H}} \bigg].$$
>
> This solution also coincides with the posterior mean of a Gaussian process regression problem. The second term in Eq. (8), $\sigma^t_h$, corresponds to the *upper confidence bound bonus* that is commonly used in the Bandit and RL literature.
>
> **Eq. (6) defines two shorthand notations for the convenience of presentation:**
> The $(t-1)\times(t-1)$ matrix $K^t_h$ is also known as the Gram matrix. The $(i,j)$-entry of matrix $K^t_h$ measures the "similarity" between the state observed and action taken in episode $i$ vs. those in episode $j$, whereas function $k^t_h(z)$ measures the "similarity" between the state-action vector $z$ vs. the state observed and action taken in time step $h$ of previous episodes.

---

> ### Author Response · Authors · 2022-11-18
> **Response to Reviewer aLWD (Part 2)**
>
> > Finally, while a smaller issue for a paper whose contributions are largely theoretical, it is a bit disappointing that after a pitch about how prior work was specialized for CVaR but this approach was more general, all the experiments only used CVaR.
>
> In Appendix E of the revised paper, we provide additional experimental results for three different risk measures, Value-at-Risk (VaR), CVaR, and Entropic Value-at-Risk (EVaR), under various risk parameters. We note that VaR is **not** a coherent risk measure, so our regret upper bound does not apply. However, we include VaR since it is a commonly used risk measure in practice and has important connections to CVaR and EVaR. We also discuss the behavior of RA-UCB and some interesting observations. For example, for the same risk parameter $\alpha$, EVaR can be more risk-averse than both CVaR and VaR at the expense of lower average rewards, which can be observed from our experimental results.
>
> We have also given details of coherent risk measures EVaR and g-entropic risk measures (which contain both CVaR and EVaR as special cases) in Appendix C of the revised paper.
>
> > I also found it hard to distinguish which parts of the analysis are novel. For example, how much of the analysis of the first term in Theorem 1 is standard?
>
> In the risk-aware RL setting, the Bellman operator is replaced by the risk-aware Bellman operator. This fundamental change invalidates all results that rely on the linearity property of the risk-neutral Bellman operator. For this reason, the analysis of Theorem 1, even for the first term, is met with several challenges and requires non-standard arguments. In the following, we would like to highlight our technical innovation in two major parts of the analysis that are crucial to the proof of Theorem 1:
>
> 1. **Regret decomposition in Lemma 1.** The standard first step to analyze RL regret bound often involves decomposing the regret into a sum of discrepancies terms over each episode $t \in [T]$ and each time step $h \in [H]$. Standard regret decomposition (Lemma 5.1 in Yang et al., 2020) relies heavily on the *linearity of expectation* property of the risk-neutral Bellman equation. This property **does not hold** for the risk-aware RL setting when the expectation in the Bellman equation is replaced with the CVaR risk measure. We overcome this challenge by leveraging the super-additivity of the CVaR risk measure. More details are in Lemma 3 in Appendix D and the paragraph that follows.
>
> 2. **Analysis to control the growth of $\xi^2_{t,h}$ (Term II of Lemma 4, defined in Eq. (17)).** In risk-neutral RL, one can show that $\\{\xi^2\_{t,h}\\}\_{t\in[T], h\in[H]}$ is a bounded martingale difference sequence and apply Azuma-Hoeffding concentration inequality. However, this sequence is no longer a martingale in the risk-aware setting. To overcome this challenge, we **design the RA-UCB algorithm** such that **the regret admits a new decomposition that allows us to control $\xi^2_{t,h}$** using the concentration inequality of the risk estimator.
> More details are in the proof of Lemma 6 in Appendix D.4.
>
> In summary, our novel technical innovations mentioned above, coupled with the algorithm design of RA-UCB, play a critical role in establishing the upper bound in Theorem 1. We believe these novel ideas will interest the community, especially in the theoretical aspects of risk-aware RL.

---

> ### Author Response · Authors · 2022-11-18
> **Response to Reviewer aLWD (Part 1)**
>
> Thank you for your detailed comments and suggestions. We appreciate your effort to understand our paper thoroughly. We have responded to each of your comments below.
>
> > The paper lacks a related work section. I was left wondering about the relationship to work such as distributional RL.
>
> We agree with the reviewer that the related work section is needed to distinguish our work from existing work. Therefore, we have moved the **related work section**, which was previously in the Appendix, to the revised main paper (see *Section 1.1*).
>
> In the related work section, we first introduce the seminal work of Howard & Matheson (1972), which is widely perceived as the first paper in risk-aware RL. We then survey recent development in the machine learning and operations research communities about various risk measures used in RL such as optimizing moments of the total reward (Jaquette, 1973), exponential utility or entropic risk (Borkar, 2001; 2002; Bäuerle & Rieder, 2014; Fei et al., 2020; 2021; Moharrami et al., 2022), mean-variance criterion (Sobel, 1982; Li & Ng, 2000; La & Ghavamzadeh, 2013; Tamar et al., 2016), and conditional value-at-risk (Boda & Filar, 2006; Artzner et al., 2007; Bäuerle & Mundt, 2009; Bäuerle & Ott, 2011; Yu et al., 2018; Rigter et al., 2021).
>
> We then narrow our attention to *closely related works* in terms of problem settings (Bäuerle & Ott, 2011; Yu et al., 2018; Fei et al., 2021; Fei & Xu, 2022) or techniques used therein (Jin et al., 2020; Yang et al., 2020). We also discuss different yet related fields such as *safe RL* and *distributional RL*.
>
> > I am somewhat concerned about the form of the objective given by equation (2). The justifications provided for it seem to center around technical convenience (e.g. guaranteeing the optimal policy is Markov). But how does this form of the objective affect the “meaning” of the risk measure chosen relative to the “static” version?
>
> The reviewer is right in the sense that our choice of the nested composition of the risk in Eq. (2) is motivated mainly by its desirable theoretical properties. However, mathematical convenience is not the sole reason for this choice.
> The formulation of dynamic risk objective has an important property known as **time consistency**. It takes a considerable amount of additional mathematical exposition to define *time consistency* formally. Due to space constraints in the main paper, we include its formal statements in Appendix B. We give a simple explanation below to allow a reader to understand the notion of *time consistency* without reading the appendix in detail.
>
> Let us start with the question: "How do we measure the risk of sequences?" Suppose that two sequences of random variables $Z_1,\dots,Z_T$ and $W_1,\dots,W_T$ are given and for $\tau \in [T]$, $Z_{\tau}=W_{\tau}$ almost surely. Then, a risk measure is time-consistent if
>
> $$\rho_T(Z_{\tau+1},\dots,Z_T) \leq \rho_T(W_{\tau+1},\dots,W_T) \Rightarrow  \rho_T(Z_{\tau},\dots,Z_T) \leq \rho_T(W_{\tau},\dots,W_T)\ .$$
>
> Intuitively, for two sequences of random variables that coincide at time $\tau$ almost surely, if we prefer $W$ over $Z$ at time $\tau + 1$, we must also prefer $W$ over $Z$ at time $\tau$. This preference implies that our risk preference stays the same over time, hence giving rise to the name *time consistency*.
>
> **Why is time consistency a desirable property**?
> This property ensures that we do not contradict ourselves in our risk evaluation. If we observe the same realization, the sequence that is better today should continue to be better tomorrow. Our risk preference stays the same over time. Note that this property is trivially satisfied in standard RL where the risk measure is replaced with expectation.
> In contrast, a single-stage risk measure (i.e., *static* version) applied on the cumulative reward $\rho\big(\sum_{h=1}^{H}r_h(x_h,a_h)\big)$ does not enjoy this *time consistency* property (Ruszczyński, 2010).
>
> We hope the above discussion sheds some light on why we use the dynamic risk measure (as defined in Eq. (2)) over the static version.

---

> ### Author Response · Authors · 2022-11-25
> **A friendly reminder**
>
> Dear Reviewer aLWD,
>
> We want to thank you for your valuable feedback. We have addressed each of your questions in the response and incorporated them in the revised manuscript.
>
> Please let us know if you have any more questions, and we would be happy to address them within our allowed period.
>
> Warmest regards,
> Authors of Paper1497.

---

> > ### Comment · Reviewer_aLWD · 2022-11-26
> > **Comments on response**
> >
> > Thank you for including related work in the main text, clarifying the technical contributions, and adding context around the choice of regret measure.

---

> > > ### Author Response · Authors · 2022-11-26
> > > **Thank you for your review**
> > >
> > > Dear Reviewer aLWD,
> > >
> > > Thank you again for acknowledging our response. Your feedback was tremendously valuable for us to improve our manuscript.
> > >
> > > Please let us know if you have further concerns/questions. If not, we sincerely hope that you can consider improving your opinion of our work.
> > >
> > > Warmest regards,
> > > Authors of Paper1497.

---

### Author Response · Authors · 2022-11-18
**General Response to all Reviewers**

We thank all reviewers for their time and efforts in evaluating our paper and for their detailed comments and suggestions. We hope our responses and answer to your questions will alleviate your concerns and further improve your opinion of our work. We have also uploaded a revised version of the paper. If you have additional questions, we would be happy to address them.

---

### Decision · Program_Chairs · 2023-01-20

**Decision:**

Accept: poster

**Justification For Why Not Higher Score:**

The lack of clarity and proper comparison with the existing results.

**Justification For Why Not Lower Score:**

The technical results of the paper are novel and seem to be solid.

**Metareview: Summary, Strengths And Weaknesses:**

The paper is on regret minimization in risk-aware finite-horizon RL with (nested) coherent risk measures and access to a simulator from which samples from the probability transition kernel can be drawn. The paper is mainly algorithmic and theoretical, supported by a simple proof-of-concept experiment. Although the reviewers appreciate the novelty of the results, there are concerns about the clarity of the presentation and the lack of proper comparison with the existing results. I personally found it surprising that there is no mention of the access to a simulator and the fact that the risk studied in the paper is nested risk and not the more challenging static risk. These are important facts about the results that should be highlighted and presented up-front. I agree with the reviewers that the authors could have done a better job in referring to the existing results. For example, the following work on coherent risk measure has not been cited:

Tamar, Chow, Ghavamzadeh, and Mannor. Policy Gradient for Coherent Risk Measures. NeurIPS-2015.

I would recommend the authors to take the reviewers' comments into account and improve their work, especially its clarity, citations, and related work, when preparing the next draft of their paper.

**Note From Pc:**

if the above contains the word "oral" or "spotlight" please see: "oral" presentation means -> notable-top-5% and "spotlight" means -> notable-top-25%. As stated in our emails, we are disassociating presentation type from AC recommendations